# Human Vγ9Vδ2 T cells exhibit antifungal activity against *Aspergillus fumigatus* and other filamentous fungi

Satoru Koga,[1,2] Takahiro Takazono,[2,3] Hodaka Namie,[3] Daisuke Okuno,[2] Yuya Ito,[2] Nana Nakada,[2,4] Tatsuro Hirayama,[2,5] Kazuaki Takeda,[2] Shotaro Ide,[2,6] Naoki Iwanaga,[2] Masato Tashiro,[3] Noriho Sakamoto,[1,2] Akira Watanabe,[7] Koichi Izumikawa,[3] Katsunori Yanagihara,[8] Yoshimasa Tanaka,[9] Hiroshi Mukae[1,2]

**ABSTRACT**  Invasive aspergillosis (IA) and mucormycosis are life-threatening diseases, especially among immunocompromised patients. Drug-resistant *Aspergillus fumigatus* strains have been isolated worldwide, which can pose a serious clinical problem. As IA mainly occurs in patients with compromised immune systems, the ideal therapeutic approach should aim to bolster the immune system. In this study, we focused on Vγ9Vδ2 T cells that exhibit immune effector functions and examined the possibility of harnessing this unconventional T cell subset as a novel therapeutic modality for IA. A potent antifungal effect was observed when *A. fumigatus* (*Af293*) hyphae were challenged by Vγ9Vδ2 T cells derived from peripheral blood. In addition, Vγ9Vδ2 T cells exhibited antifungal activity against hyphae of all *Aspergillus* spp., *Cunninghamella bertholletiae*, and *Rhizopus microsporus* but not against their conidia. Furthermore, Vγ9Vδ2 T cells also exhibited antifungal activity against azole-resistant *A. fumigatus*, indicating that Vγ9Vδ2 T cells could be used for treating drug-resistant *A. fumigatus*. The antifungal activity of Vγ9Vδ2 T cells depended on cell-to-cell contact with *A. fumigatus* hyphae, and degranulation characterized by CD107a mobilization seems essential for this activity against *A. fumigatus*. Vγ9Vδ2 T cells could be developed as a novel modality for treating IA or mucormycosis.

**IMPORTANCE** Invasive aspergillosis (IA) and mucormycosis are often resistant to treatment with conventional antifungal agents and have a high mortality rate. Additionally, effective antifungal treatment is hindered by drug toxicity, given that both fungal and human cells are eukaryotic, and antifungal agents are also likely to act on human cells, resulting in adverse effects. Therefore, the development of novel therapeutic agents specifically targeting fungi is challenging. This study demonstrated the antifungal activity of Vγ9Vδ2 T cells against various *Aspergillus* spp. and several *Mucorales in vitro* and discussed the mechanism underlying their antifungal activity. We indicate that adoptive immunotherapy using Vγ9Vδ2 T cells may offer a new therapeutic approach to IA.

**KEYWORDS**  γδ T cell, invasive aspergillosis, filamentous fungi, nitrogen-containing bisphosphonate prodrug

**Ad Hoc Peer Reviewer** Sebastian Thomas Wurster

Address correspondence to Takahiro Takazono, takahiro-takazono@nagasaki-u.ac.jp.

Y.T. is a co-inventor of JP2014-257451 (on the method for expanding γδ T cells using PTA) and JP2014-73475 (on the method for a nonradioactive cellular cytotoxicity assay). The other authors declare that the research was conducted without any commercial or financial relationships that could be construed as a potential conflict of interest.

See the funding table on p. 15.

*A*spergillus molds, especially *Aspergillus fumigatus*, are causative agents of invasive aspergillosis (IA), a fatal disease occurring mainly in severely immunocompromised hosts, such as patients with hematologic malignancies and transplant recipients (1–3). Despite the development of novel antifungal agents and improved treatment strategies, the mortality rate of IA remains high at 35% (1). Furthermore, the increase and global spread of azole-resistant *A. fumigatus* hampers conventional IA treatments and poses a worldwide challenge (4–7). In addition, mucormycosis is an invasive fungal infection

caused by over 200 species, including *Rhizopus* spp. and *Cunninghamella* spp. (8). These fungal species are naturally resistant to azole and other antifungals, contributing to a high fatality rate of 54% (9).

Recent research advances in the mechanism underlying immune responses to *Aspergillus* spp. (10–12) indicate that IA resistance to standard therapy may result from host's failure to induce appropriate immune responses. Indeed, the outcome of invasive mold infections in severely immunocompromised patients depends on host factors, including the resolution of neutropenia (13, 14). This has led to the emergence of immunotherapy as an alternative approach to conventional antifungal drug treatment (15). The antifungal effects of natural killer (NK) cells against *A. fumigatus* have been demonstrated, and the adoptive transfer of NK cells for treating IA has been explored (16–18).

Besides NK cells, innate or innate-like immune effector cells exist in humans, such as CD4CD8-double negative T cells, including NKT cells and γδ T cells. Human Vγ9Vδ2-bearing γδ T cells (Vγ9Vδ2 T cells) account for 1%–5% of circulating T cells in the peripheral blood, exhibit innate immune-like functions, and can damage infected cells and tumor cells similar to NK cells (19, 20). In a previous study, Vγ9Vδ2 T cells produced a significant level of tumor necrosis factor-α (TNF-α) in response to water-soluble extracts from *Aspergillus* spp. (21); however, the antigen and mechanism underlying the antigen recognition remain unknown. It is also unknown whether Vγ9Vδ2T cells possess antifungal activity against *Aspergillus*.

In healthy adults, the majority of peripheral blood γδ T cells express Vγ9Vδ2-bearing T cell receptors (TCRs), which recognize microbial (E)-4-hydroxy-3-methyl-but-2-enyl pyrophosphate (HMBPP) in the 2-*C*-methyl-D-erythritol 4-phosphate/1-deoxy-D-xylulose 5-phosphate (MEP/DOXP) pathway and self-isopentenyl diphosphate (IPP) and dimethylallyl diphosphate (DMAPP) in the mevalonate pathway in a butyrophilin (BTN) 2A1/3A1-dependent manner (22, 23).

HMBPP is produced by some pathogenic bacteria, such as *Mycobacterium tuberculosis*, *Mycobacterium bovis*, *Listeria monocytogenes*, *Escherichia coli*, *Salmonella typhimurium*, and parasites, such as *Plasmodium falciparum* and *Toxoplasma gondii* (24, 25). When Vγ9Vδ2 T cells recognize these pathogen-derived metabolites, they readily proliferate and produce interferon-γ (IFN-γ) and TNF-α (26), mounting a rapid response against the pathogens. The antibacterial activity of Vγ9Vδ2 T cells against *M. tuberculosis* has been reported (27, 28). A clinical study on allogeneic Vγ9Vδ2 T cell-based immunotherapy in patients with multidrug-resistant tuberculosis demonstrated the regimen to be well tolerated and effective against the pathogen (29), indicating its potential applicability to IA.

We have previously shown that Vγ9Vδ2 T cells from peripheral blood could be expanded *ex vivo* using PTA, a nitrogen-containing bisphosphonate prodrug and an inhibitor of farnesyl diphosphate synthase (FDPS), and interleukin-2 (IL-2) (30, 31). In addition, a clinical trial of therapeutic administration of Vγ9Vδ2T cells to patients with malignant tumors has revealed that the regimen is well tolerated (32). In this study, we evaluated the antifungal activity of *ex vivo* expanded/activated human Vγ9Vδ2T cells against *Aspergillus* spp. and other *Mucorales in vitro* and explored the mechanism underlying their antifungal activity. Our findings will help develop Vγ9Vδ2 T cells as a novel treatment modality for IA or mucormycosis.

## RESULTS

### Human Vγ9Vδ2 T cells exhibit direct cytotoxic activity against *A. fumigatus* filamentous hyphae

To examine whether human Vγ9Vδ2 T cells exhibit antifungal activity against *A. fumigatus*, we assessed their effect on the viability of *A. fumigatus* hyphae in a direct co-culture system. We first expanded Vγ9Vδ2 T cells from peripheral blood mononuclear cells (PBMCs) derived from healthy adult volunteers using PTA and IL-2. The initial proportion of CD3$^+$Vδ2$^+$ T cells in PBMCs was 10.7% (Fig S1A). After expansion with

PTA/IL-2, the proportion of CD3$^+$Vδ2$^+$ T cells reached 99.5%, with almost all the CD3$^+$Vδ2$^+$ T cells expressing Vγ9. The number of Vγ9Vδ2 T cells increased by 1,008-fold, increasing from $1.31 \times 10^6$ cells to $1.32 \times 10^9$ cells. During the expansion with PTA/IL-2, Vγ9Vδ2 T cells began to form explicit clusters on day 4, and maximum clustering was attained on day 6. On day 11, the resulting highly homogeneous Vγ9Vδ2 T cells were frozen and stored in a liquid nitrogen tank until used for subsequent assays.

To confirm the effector functions of the PTA/IL-2-expanded Vγ9Vδ2 T cells, we first examined the cell surface markers expressed on Vγ9Vδ2 T cells. As shown in Fig S2, the PTA/IL-2-expanded Vγ9Vδ2 T cells expressed a high level of NKG2D (CD314), an activating C-type lectin-like receptor, and DNAX accessory molecule-1 (DNAM-1, CD226), a type I membrane protein containing 2 Ig-like C2-type domains. However, the expression levels of FasL, a Fas ligand (CD95L), and TRAIL, a 30 kDa transmembrane protein known as TNF-related apoptosis-inducing ligand (CD253, also called TNFSF10 or Apo-2L), were marginal. This indicates that Vγ9Vδ2 T cells exert effector functions through NKG2D and DNAM-1 but not cell death receptor ligands. When Raji, a human malignant Burkitt's lymphoma cell line, was incubated with the PTA/IL-2-expanded Vγ9Vδ2 T cells, over 30% of Raji cells were killed by Vγ9Vδ2 T cells at an effect-to-target (E/T) ratio of 200, in which Vγ9Vδ2 T cells exhibited the anti-tumor effect in an E/T ratio-dependent manner. Furthermore, the addition of PTA to this system further enhanced the effector functions, with Vγ9Vδ2 T cells killing approximately 80% of Raji cells in 1 h at the E/T ratio of 200, confirming that the PTA/IL-2-expanded Vγ9Vδ2 T cells could exhibit potent effector functions against tumor cells(online supplementary file 1).

In addition, we examined several cell surface makers expressed on Vγ9Vδ2 T cells derived from peripheral blood stimulated with PTA/IL-2 for 0, 3, and 11 days to gain insight into the possible functions of Vγ9Vδ2 T cells in stationary and activated phases. As shown in Fig S4, the expression of CD369 (Dectin-1, CLEC7A), CD282 (TLR-2), CD284 (TLR-4), FasL, and TRAIL was low at all the time points. In contrast, a high level of DNAM-1 expression was observed at all time points. The expression of NKG2D was initially high but decreased on day 3 after stimulation and subsequently resumed to an initial level on day 11. Conversely, CD25 was not expressed in the steady state, but was expressed to a markedly high level on day 3 after stimulation, and the expression gradually decreased thereafter. Programmed death-1 (PD-1, CD279) was slightly expressed on day 0, and the expression level increased on day 3 and returned to the stationary state on day 11.

We subsequently examined the effect of Vγ9Vδ2 T cells on the hyphae of *A. fumigatus*. When *A. fumigatus* filamentous hyphae were cultured in the presence of Vγ9Vδ2 T cells, the viability of *A. fumigatus* was significantly reduced, compared to that in the absence of Vγ9Vδ2 T cells, strongly indicating that Vγ9Vδ2 T cells have potent antifungal activity, which was comparable to that of NK cells (Fig. 1A). Notably, Vγ9Vδ2 T cells exhibit the antifungal activity in a Vγ9Vδ2 T cell number-dependent manner, and antifungal activity of Vγ9Vδ2 T cells after cryopreservation was comparable to that before cryopreservation (Fig. 1A).

When other filamentous fungi species were examined in the same assay, Vγ9Vδ2T cells exhibited similar antifungal activity to all the other *Aspergillus* species, *C. bertholletiae*, and *R. microsporus* (Fig. 1B), but not to *Rhizopus oryzae*, *Rhizomucor pusillus,* and *Mucor circinelloides*.

## Vγ9Vδ2 T cells exhibit antifungal activity against azole-resistant and susceptible *A. fumigatus* strains

We examined the antifungal effect of Vγ9Vδ2 T cells against azole-resistant *A. fumigatus* strains with different resistance mechanisms (*NGS-ER7, MF2108, IFM63240*). When azole-resistant *A. fumigatus* strains (hyphae-form) were co-cultured with Vγ9Vδ2 T cells, the Vγ9Vδ2 T cells exhibited potent antifungal activity against these fungi to a similar level as with the azole-susceptible *A. fumigatus* strain (Fig. 2). The complex intertwining hyphal layers were observed under an optical microscope after conidial suspensions were incubated for 24 h on a polystyrene plate. We subsequently compared the antifungal

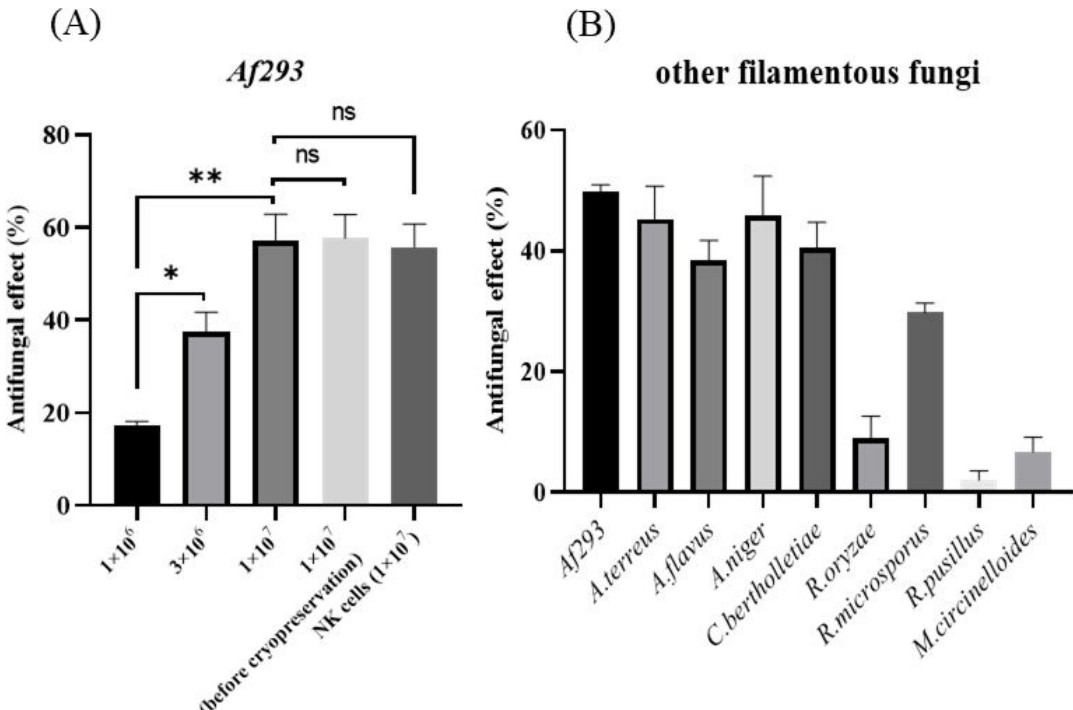

**FIG 1** Effect of Vγ9Vδ2 T cells on the viability of filamentous fungi. (A) Dose-dependent antifungal activity of Vγ9Vδ2 T cells against *Af293*, a wild-type *A. fumigatus* strain, which was comparable to that of NK cells, and cryopreservation did not significantly alter the antifungal activity of Vγ9Vδ2 T cells. *Af293* (*A. fumigatus*) was challenged by $1 \times 10^6$, $3 \times 10^6$, or $1 \times 10^7$ Vγ9Vδ2 T cells, and the antifungal activity was determined ($*P = 0.0012$; $**P = 0.0003$). (B) Species specificity in the antifungal effect of Vγ9Vδ2 T cells on *Af293*. *Af293*, *Aspergillus terreus*, *Aspergillus flavus*, *Aspergillus niger*, *C. bertholletiae*, *Rhyzopus oryzae*, *R. microsporus*, *Mucor circinelloides*, and *Rhizomucor pusillus* were challenged by Vγ9Vδ2 T cells, and the antifungal effect was observed. Data shown are representative of at least three independent experiments.

effect of Vγ9Vδ2 T cells and voriconazole against *A. fumigatus* forming the hyphal layers. Vγ9Vδ2 T cells exerted antifungal activity against *A. fumigatus*-hyphae even in the presence of the hyphal layers. Alternatively, all species of *A. fumigatus*, including azole-susceptible and several azole-resistant strains, were resistant to voriconazole on the condition of forming the hyphal layer (Fig. 2). Based on these findings, Vγ9Vδ2 T cells could potentially be developed as a novel antifungal modality for the treatment of azole-resistant *A. fumigatus* and wild-type strains. In addition, Vγ9Vδ2 T cells are effective against *A. fumigatus* forming hyphal layers resistant to antifungal agents.

## *A. fumigatus*-conidia are resistant to the antifungal activity of Vγ9Vδ2 T cells

In previous studies, *A. fumigatus*-conidia did not exhibit immunogenicity in some human effector cells (17, 29). Therefore, we examined the effect of Vγ9Vδ2 T cells on the viability of *A. fumigatus*-conidia. When *A. fumigatus*-conidia were cultured in the presence of Vγ9Vδ2 T cells, the viability of *A. fumigatus*-conidia remained unaltered judging from the results of the standard colony-forming units (CFUs) assay, indicating that Vγ9Vδ2 T cells failed to recognize the conidial form of *A. fumigatus* (Fig. 3A), in contrast to *A. fumigatus*-hyphae. Vγ9Vδ2 T cells likely recognize hyphae, an invasive form of *A. fumigatus*, but not conidia, an inert form of the fungi.

## Vγ9Vδ2 T cells exert antifungal activity against an *A. fumigatus* mutant strain which is deficient of galactomannan

To identify the target for the recognition of *A. fumigatus* by Vγ9Vδ2 T cells, we employed *Δuge5*, a mutant strain of *A. fumigatus*, defective in galactomannan (GM) in the cell wall.

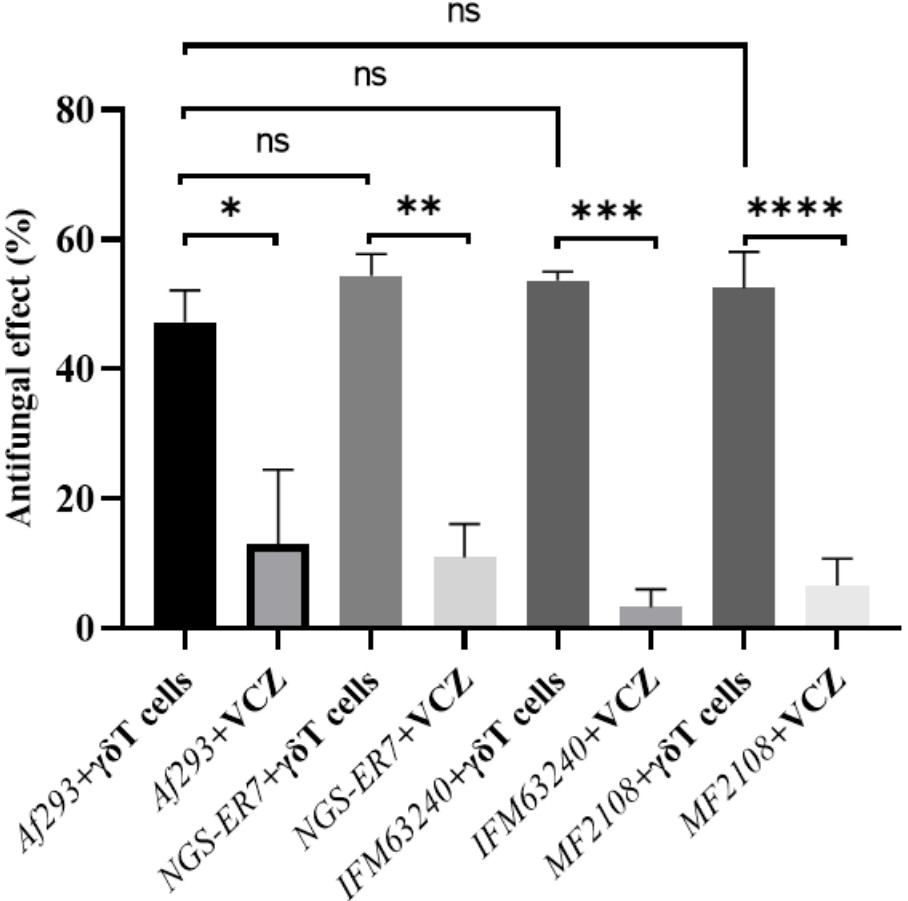

**FIG 2** Effect of Vγ9Vδ2 T cells on the viability of azole-resistant *A. fumigatus*. Some types of azole-resistant *A. fumigatus* strains were challenged by Vγ9Vδ2 T cells, and the antifungal activity was determined in the presence or absence of voriconazole (*$P$ = 0.0089; **$P$ = 0.0003; ***$P$ < 0.0001; ****$P$ = 0.0003). Data shown are representative of at least three independent experiments.

The antifungal activity against Δ*uge5* strain was examined using the standard 2,3-bis[2-methoxy-4-nitro-5-sulphenyl]-2*H*-tetrazolium-5-carboxyanilide sodium salt (XTT) assay. As shown in Fig. 3B, the effect of Vγ9Vδ2 T cells on the viability of the Δ*uge5* strain was similar to that on the wild-type *A. fumigatus* strain, indicating GM is not essential for the recognition of *A. fumigatus* by Vγ9Vδ2 T cells.

## *A. fumigatus*-hyphae suppresses the secretion of both IFN-γ and TNF-α from Vγ9Vδ2 T cells

Th1-type immune responses play a pivotal role in protective immunity against IA (10). It has been reported that Vγ9Vδ2 T cells could produce a significant level of TNF-α in response to water-soluble extracts of *A. fumigatus* (21). Therefore, we examined the levels of IFN-γ and TNF-α in the culture supernatants of living *A. fumigatus*-hyphae and Vγ9Vδ2 T cells. Living *A. fumigatus*-hyphae failed to induce the secretion of IFN-γ and TNF-α from Vγ9Vδ2 T cells (Fig. 4A and C). To examine whether the PTA/IL-2-expanded Vγ9Vδ2 T cells can secrete IFN-γ and TNF-α in response to antigenic stimuli through TCR, we treated Vγ9Vδ2 T cells with HMBPP, which is known to stimulate Vγ9Vδ2 T cells in a Vγ9Vδ2-bearing TCR-dependent manner (26). As shown in Fig. 4B and D, HMBPP-stimulated Vγ9Vδ2 T cells secreted a significant level of IFN-γ and TNF-α, demonstrating that the PTA/IL-2-expanded Vγ9Vδ2 T cells had the ability to secrete IFN-γ and TNF-α in response to TCR stimuli.

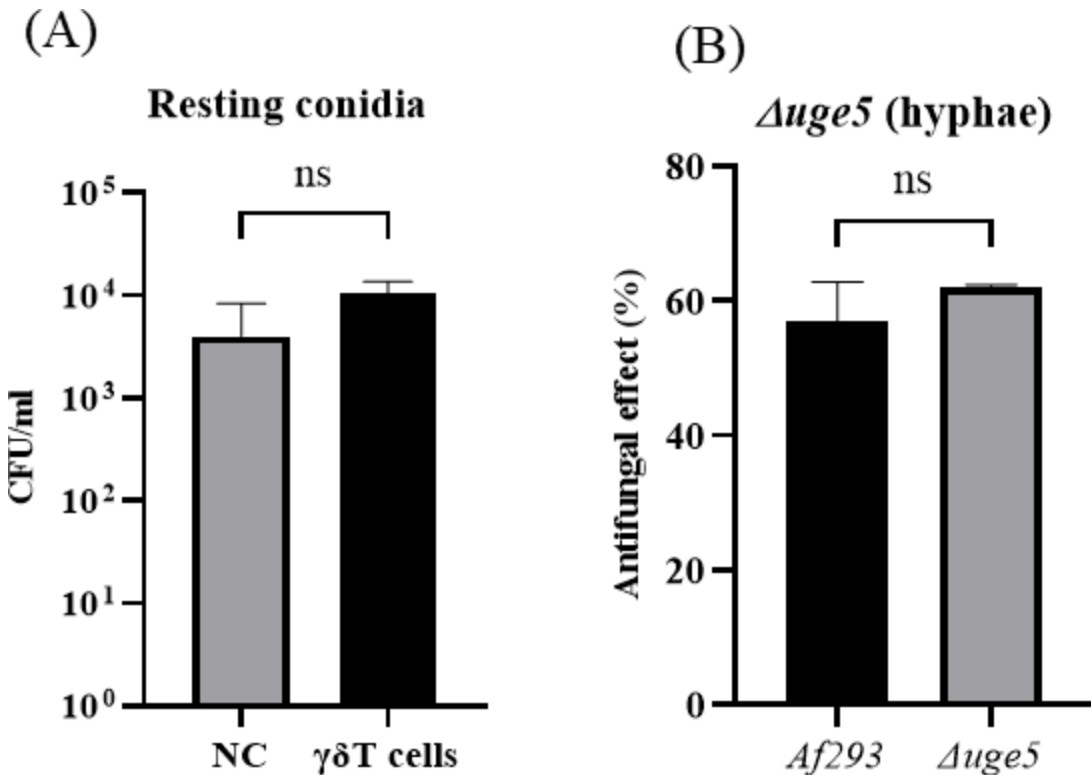

**FIG 3** Comparison of the effects of the morphological form and metabolic alteration of *A. fumigatus* on the antifungal activity of Vγ9Vδ2 T cells. (A) Effect of Vγ9Vδ2 T cells on the viability of *A. fumigatus*-conidia. Af293 (*A. fumigatus*)-conidia were challenged by Vγ9Vδ2 T cells, and the antifungal activity was determined. (B) Effect of deficiency in galactomannan on the antifungal activity of Vγ9Vδ2 T cells against *A. fumigatus*. *Δuge5*, an *A. fumigatus*-mutant defective in galactomannan (GM), was challenged by Vγ9Vδ2 T cells, and the antifungal activity was determined. Data shown are representative of at least three independent experiments.

Subsequently, we examined the effect of *A. fumigatus*-hyphae on the secretion of these cytokines from HMBPP-stimulated Vγ9Vδ2 T cells. Surprisingly, the addition of *A. fumigatus*-hyphae markedly decreased both IFN-γ and TNF-α secretion from Vγ9Vδ2 T cells in response to HMBPP (Fig. 4B and D). To examine the effect of *A. fumigatus*-hyphae on the viability of Vγ9Vδ2 T cells, Vγ9Vδ2 T cells were cocultured in the presence or absence of *A. fumigatus*-hyphae, and the number of viable Vγ9Vδ2 T cells was counted. The standard trypan blue dye exclusion demonstrated no difference in the number of viable Vγ9Vδ2 T cells between the two groups, indicating that the secretion of both IFN-γ and TNF-α from HMBPP-stimulated Vγ9Vδ2 T cells was significantly suppressed by the presence of *A. fumigatus*-hyphae.

**Antifungal activity of Vγ9Vδ2 T cells against *A. fumigatus* is contact-dependent**

Based on the aforementioned findings, it is evident that Vγ9Vδ2 T cells could exert antifungal effects. Subsequently, we examined the mechanism underlying the immune effector functions of Vγ9Vδ2 T cells against *A. fumigatus*. To examine the effect of cell-to-cell contact on the antifungal activity of Vγ9Vδ2 T cells, we employed a cell culture insert system comprising a 0.4 µm pore membrane insert. After culturing Vγ9Vδ2 T cells and *A. fumigatus*-hyphae using the culture insert, the viability of *A. fumigatus*-hyphae in this system was significantly higher than that in the co-culture system that allowed cell-to-cell contact between Vγ9Vδ2 T cells and *A. fumigatus*-hyphae (Fig. 4E), indicating that cell-to-cell contact is essential for the antifungal activity of Vγ9Vδ2 T cells. In addition, culture supernatants derived from the co-culture system of Vγ9Vδ2 T cells and *A.*

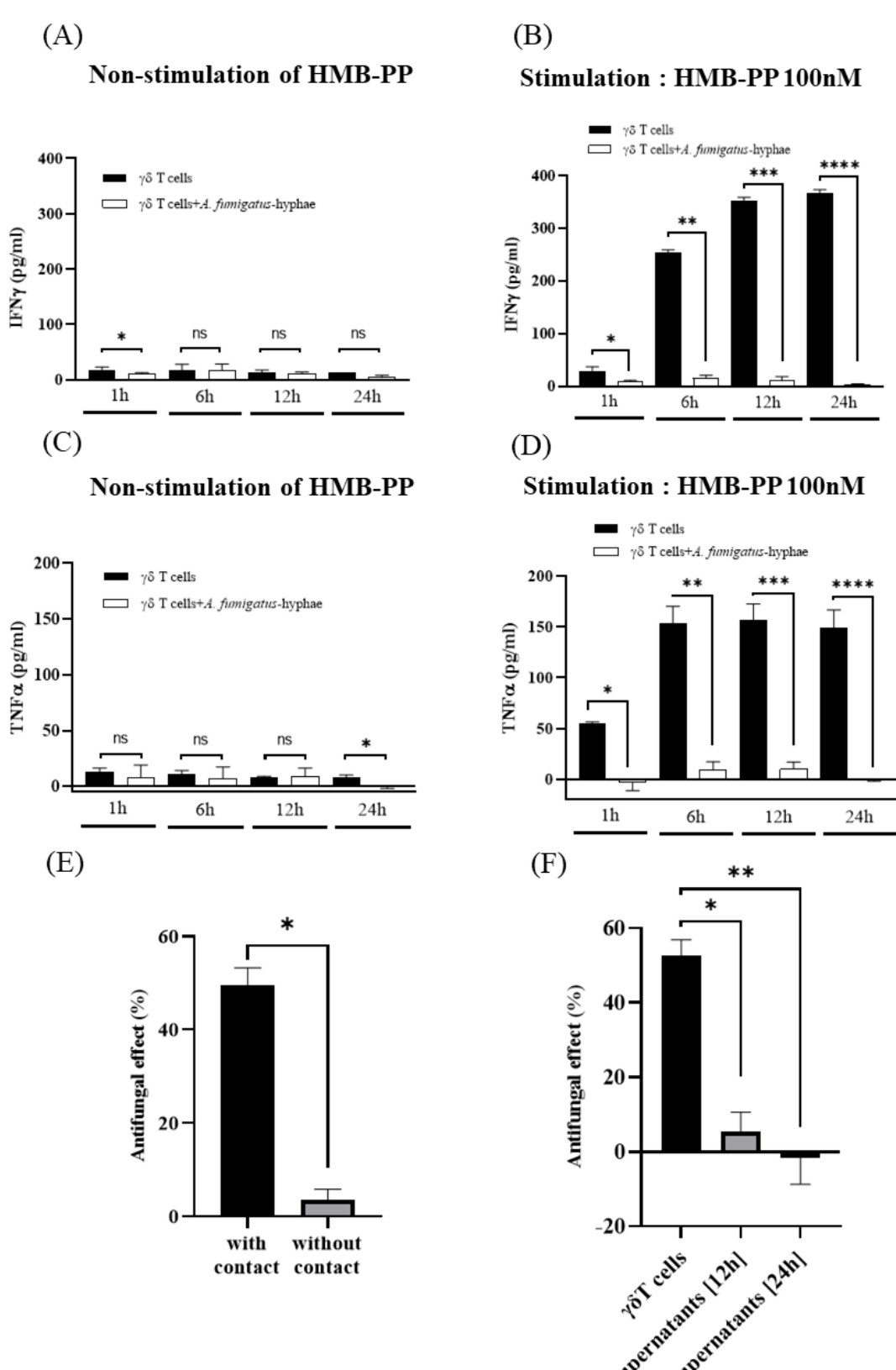

**FIG 4** Analyses of the mechanism underlying the antifungal activity of Vγ9Vδ2 T cells against *A. fumigatus*. (A) Effect of *A. fumigatus*-hyphae on the secretion of IFN-γ by Vγ9Vδ2 T cells. Vγ9Vδ2 T cells were treated with *Af293* (*A. fumigatus*)-hyphae, and the culture supernatants were examined for IFN-γ (*$P = 0.042$). (B) Effect of *A. fumigatus*-hyphae on the secretion of IFN-γ by Vγ9Vδ2 T cells stimulated with HMBPP. Vγ9Vδ2 T cells were treated with HMBPP and *Af293* (Continued on next page)

**FIG 4** (Continued)

(*A. fumigatus*)-hyphae, and the culture supernatants were examined for IFN-γ (*P = 0.0037; **P < 0.0001; ***P < 0.0001; ****P < 0.0001). (C) Effect of *A. fumigatus*-hyphae on the secretion of TNF-α by Vγ9Vδ2 T cells. Vγ9Vδ2 T cells were treated with *Af293* (*A. fumigatus*)-hyphae, and the culture supernatants were examined for TNF-α (*P = 0.0014). (D) Effect of *A. fumigatus*-hyphae on the secretion of TNF-α by Vγ9Vδ2 T cells stimulated with HMBPP. Vγ9Vδ2 T cells were treated with HMBPP and *Af293* (*A. fumigatus*)-hyphae, and the culture supernatants were examined for TNF-α (*P < 0.0001; **P < 0.0001; ***P < 0.0001; ****P < 0.0001). (E) Effect of direct interaction between *A. fumigatus*-hyphae and Vγ9Vδ2 T cells on the antifungal activity of Vγ9Vδ2 T cells. Vγ9Vδ2 T cells and *Af293* (*A. fumigatus*)-hyphae were cultured using a cell insert culture system, and the antifungal activity was determined (*P < 0.0001). (F) Effect of supernatants from the culture of Vγ9Vδ2 T cells and *A. fumigatus*-hyphae on the antifungal activity of Vγ9Vδ2 T cells. *Af293* (*A. Fumigatus*) was challenged by co-culture supernatants of Vγ9Vδ2 T cells and *Af293* (*A. fumigatus*)-hyphae, and the antifungal activity was determined (*P = 0.0002; **P = 0.0003). Data shown are representative of at least three independent experiments.

*fumigatus*-hyphae demonstrated no antifungal activity against *A. fumigatus*-hyphae (Fig. 4F), further corroborating the hypothesis that a direct contact between Vγ9Vδ2 T cells and *A. fumigatus*-hyphae is required for the antifungal activity of Vγ9Vδ2 T cells against *A. fumigatus*-hyphae.

## *A. fumigatus*-hyphae induced degranulation in Vγ9Vδ2 T cells

The degranulation of intracellular vesicles containing effector molecules is a major mechanism by which immune effector cells exhibit contact-dependent lytic activity against target cells. As CD107a is expressed in intracellular vesicles and translocated onto cell surface membranes during degranulation, the upregulation of CD107a on cell surface membranes can be used as a marker for the degree of degranulation of immune effector cells (33). Therefore, we examined the levels of CD107a expression on Vγ9Vδ2 T cells after incubation with *A. fumigatus*-hyphae or culture supernatants derived from the co-culture system of Vγ9Vδ2 T cells and *A. fumigatus*-hyphae. As shown in Fig. 5A and B, the expression of CD107a on Vγ9Vδ2 T cells was significantly upregulated when Vγ9Vδ2 T cells were co-cultured with *A. fumigatus*-hyphae, compared to those treated with supernatants of the co-culture. Notably, the upregulation of CD107a on Vγ9Vδ2 T cells was dependent on the dose of *A. fumigatus*-hyphae (Fig. 5A and B), confirming that the cell-to-cell contact between Vγ9Vδ2 T cells and *A. fumigatus*-hyphae is essential for the antifungal activity of Vγ9Vδ2 T cells.

## Antifungal activity of Vγ9Vδ2T cells against *A. fumigatus* is mediated by calcium ($Ca^{2+}$)-dependent degranulation

CD107a-positive intracellular vesicles contain lytic molecules, such as granulysin, perforin, and granzyme B, and the release of these molecules required for their effector functions is $Ca^{2+}$-dependent (34, 35). Ethylene glycol-bis(2-aminoethylether)-*N,N,N′,N′*-tetra acetic acid A (EGTA) inhibits degranulation owing to its calcium-chelating effect (36, 37). To examine the effect of degranulation in the antifungal activity of Vγ9Vδ2 T cells, we evaluated the antifungal activity of Vγ9Vδ2 T cells after pretreatment with EGTA. As illustrated in Fig. 5C, the antifungal activity of Vγ9Vδ2 T cells against *A. fumigatus* was significantly suppressed by the pretreatment with EGTA, strongly indicating that degranulation was inexorably linked to the antifungal activity of Vγ9Vδ2 T cells.

## DISCUSSION

In this study, we performed *in vitro* antifungal effects of Vγ9Vδ2 T cells against *A. fumigatus* and other filamentous fungi that could cause invasive pulmonary infections and explored possible underlying mechanisms. Vγ9Vδ2 T cells exhibited significant antifungal effects against many species of filamentous fungi-hyphae, including azole-susceptible and resistant *A. fumigatus,* and *Mucorales* such as *C. bertholletiae* and *R. microsporus*. However, Vγ9Vδ2 T cells did not exhibit antifungal effects against the resting conidia of *A. fumigatus*. In addition, the antifungal effect of Vγ9Vδ2 T cells depended on direct contact between Vγ9Vδ2 T cells and fungi. Direct cell-to-fungus contact upregulated the expression of CD107a, a marker of degranulation, on the surface

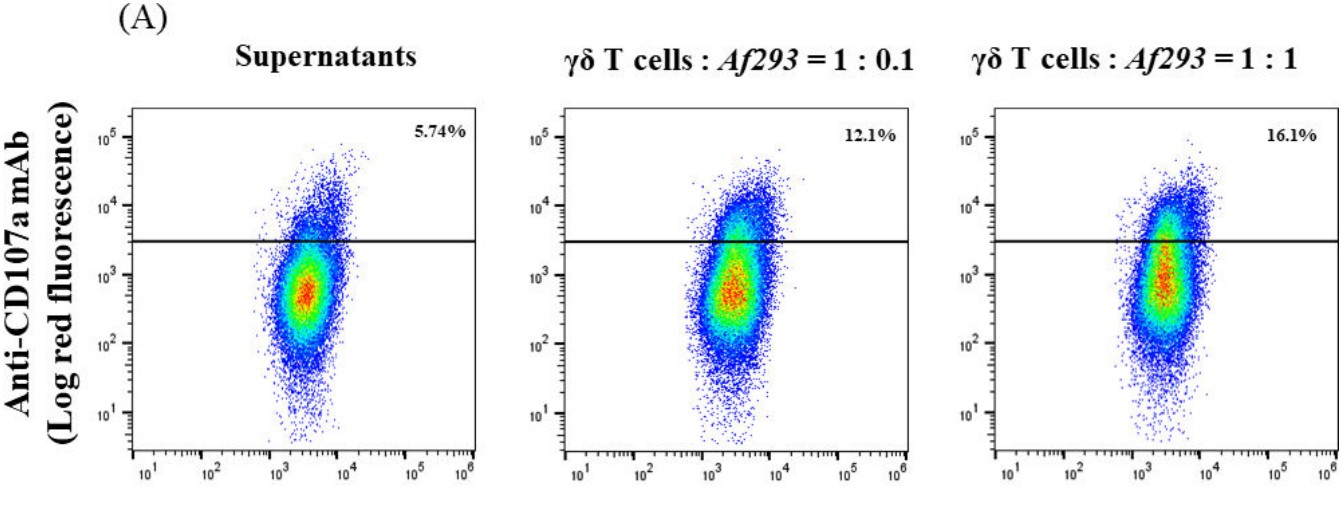

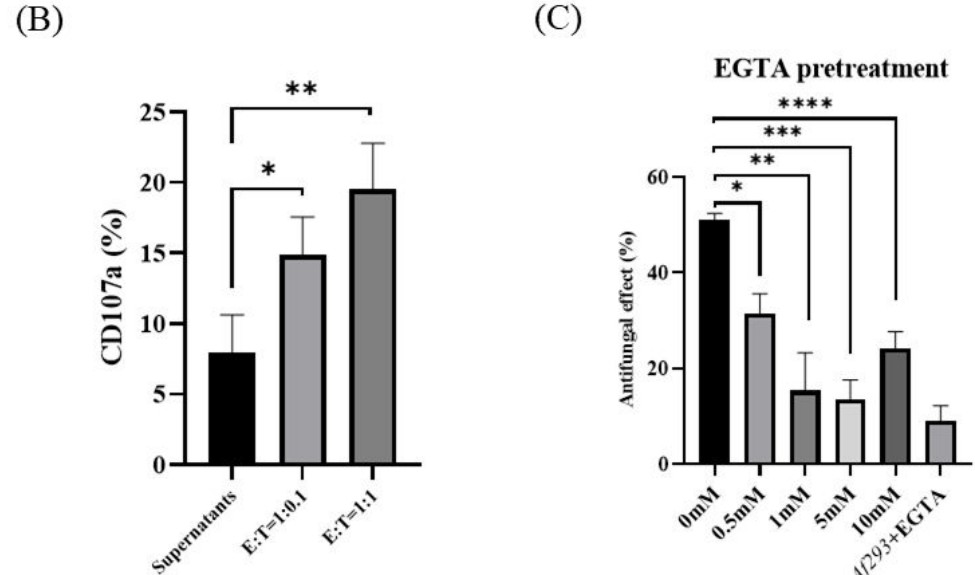

**FIG 5** Involvement of degranulation in the antifungal effect of Vγ9Vδ2 T cells against *A. fumigatus*-hyphae. (A, B) Analyses of degranulation after treatment of Vγ9Vδ2 T cells with *A. fumigatus*-hyphae. Vγ9Vδ2 T cells were treated with *Af293* (*A. fumigatus*)-hyphae or culture supernatants derived from the co-culture system of Vγ9Vδ2 T cells and *A. fumigatus*-hyphae, and the degranulation in Vγ9Vδ2 T cells was monitored through CD107a mobility on a flow cytometric analysis (*$P$ = 0.0331; **$P$ = 0.0084). (C) Effect of a $Ca^{2+}$-chelating inhibitor on the antifungal activity of Vγ9Vδ2 T cells against *A. fumigatus*. Vγ9Vδ2 T cells were challenged by *A. fumigatus*-hyphae in the presence or absence of EGTA, a $Ca^{2+}$-chelating inhibitor, and the antifungal activity of Vγ9Vδ2 T cells against *A. fumigatus*-hyphae was examined (*$P$ = 0.0014; **$P$ = 0.0015; ***$P$ = 0.0001; ****$P$ = 0.0003). Data shown are representative of at least three independent experiments.

of Vγ9Vδ2 T cells. Inhibition of degranulation with EGTA pretreatment greatly impaired the antifungal effects of Vγ9Vδ2 T cells. In addition, *A. fumigatus*-hyphae strongly inhibited the release of IFN-γ and TNF-α from Vγ9Vδ2 T cells.

We previously synthesized PTA, a nitrogen-containing bisphosphonate prodrug, and demonstrated that Vγ9Vδ2 T cells were efficiently expanded *in vitro* by PTA and IL-2. Since PTA is highly hydrophobic, the prodrug is spontaneously internalized into immune cells, such as dendritic cells and monocytes, where it is hydrolyzed by intracellular esterases to yield 2-(thiazole-2-ylamino) ethylidene-1,1-bisphosphonate (TA), an active form of PTA (38, 39). The resulting TA inhibits intracellular FDPS, which catalyzes the

formation of geranyl diphosphate from IPP and DMAPP and that of farnesyl diphosphate from geranyl diphosphate and IPP in the isoprenoid synthetic pathway (31). Inhibition of FDPS by TA results in the intracellular accumulation of IPP and DMAPP, which interact with the B30.2 intracellular domain of BTN3A1 (40). Finally, Vγ9Vδ2 T cells subsequently recognize the PTA-pulsed antigen-presenting cells within the context of BTN2A1/3A1. However, the precise recognition mechanism has not been fully elucidated (22, 41).

Vγ9Vδ2 T cells were successfully expanded by PTA/IL-2 within 11 days, reaching a purity of up to 99.5% without any purification procedure. In this study, we used the highly purified Vγ9Vδ2 T cells and demonstrated their potent cytolytic activity against Raji, a human Burkitt's lymphoma cell line. As expected, Vγ9Vδ2 T cells also exhibited palpable antifungal activity against *A. fumigatus*, a causative agent of IA, and other fungi, such as *A. terreus*, *A. flavus*, *A. niger*, *C. bertholletiae*, and *R. microsporus*. The antifungal activity of Vγ9Vδ2 T cells against *A. fumigatus* was comparable to that of NK cells. The advantage of Vγ9Vδ2 T cells over NK cells is that the expansion rate of Vγ9Vδ2 T cells is much greater than that of NK cells. When PBMCs are stimulated with PTA/IL-2, a large number of highly purified Vγ9Vδ2 T cells can be obtained within 11 days without purification, indicating that Vγ9Vδ2 T cell-based adoptive transfer therapy is more practical than that of NK cells.

In addition, Vγ9Vδ2 T cells also exhibited antifungal effects against some azole-resistant *A. fumigatus* strains with different resistance mechanisms (*NGS-ER7, MF2108, IFM63240*), strongly indicating that Vγ9Vδ2 T cell-based immunotherapy could serve as an alternative modality for the treatment of IA caused by wild-type and azole-resistant *A. fumigatus*. Since common mutations in *A. fumigatus* responsible for drug resistance are generally unrelated to the structure and components of cell walls that are highly antigenic for the host immune system, the antifungal activity of Vγ9Vδ2 T cells is not affected by such azole-resistant mutations. In addition, Vγ9Vδ2 T cells were effective against *A. fumigatus* forming complex intertwining hyphal layers resistant to antifungal agents. Notably, it was reported that *Aspergillus* hyphae formed hyphal layers with numerous mycelia intertwined on a polystyrene plate, and the mycelia within the structures were more resistant to antifungal agents including voriconazole than planktonic ones (42, 43).

The existence of azole-resistant *A. fumigatus* has been reported worldwide (44), posing a clinically significant problem in future (7, 45). In addition, biofilm-forming *A. fumigatus* has been observed in IA (46) and chronic pulmonary aspergillosis, which are resistant to conventional antifungals (33). Therefore, it is imperative to develop Vγ9Vδ2 T cell-based immunotherapy, which might overcome drug resistance of *A. fumigatus* or other filamentous fungi.

Whereas various pathogenic bacteria produce HMBPP (47), *A. fumigatus* does not have the MEP/DOXP pathway and, therefore, does not produce HMBPP, indicating that Vγ9Vδ2 T cells recognize antigenic entities other than HMBPP. Therefore, we employed a mutant *A. fumigatus* strain to examine the effect of gene mutations on the antifungal activity of Vγ9Vδ2 T cells. In this study, we used an *A. fumigatus* mutant strain, *Δuge5*, which is defective in GM, and found that Vγ9Vδ2 T cells exhibited antifungal effects against GM-deficient *A. fumigatus* as the wild-type *A. fumigatus* (*Af293*). The finding indicates that GM might not be the target for the recognition of *A. fumigatus* by Vγ9Vδ2 T cells. In addition, we examined Vγ9Vδ2 T cells for the expression of several cell surface markers, including PRRs (Dectin-1, TLR2, and TLR4), which have been reported to be associated with *Aspergillus* infections (48), NK cell-related markers, NKG2D and DNAM-1, cell death receptor ligands (FasL and TRAIL), CD25, an activation marker of Vγ9Vδ2 T cells, and PD-1, a T cell exhaustion marker, on days 0, 3, and 11 after stimulation with PTA/IL-2. Dectin-1 is one of the most well-characterized PRRs recognizing β-D-glucans (48, 49), whereas its expression on Vγ9Vδ2 T cells had not been reported. It was demonstrated that TLR2 and TLR4 were upregulated on Vγ9Vδ2 T cells after stimulation with TLR4 ligands (50), leading to induced antibacterial responses (51). In this study, PRRs were not expressed on Vγ9Vδ2 T cells under a condition we employed. In addition, the expression

levels of other cell surface markers remained noticeably unchanged before and after the expanded culture, except for NKG2D, CD25, and PD-1.

Vγ9Vδ2 T cells exhibited antifungal activity against *A. fumigatus*-hyphae but not conidia. This finding is similar to previous reports indicating that dendritic cells, alveolar macrophages, and NK cells were not activated by resting conidia (17, 52). The hydrophobic layer of *A. fumigatus*-conidia disappears during germination and transformation into hyphae. Notably, the conidial morphotype of *ΔrodA* mutant, a strain genetically deficient in the hydrophobic layer, activates dendritic cells and alveolar macrophages, indicating that the hydrophobic layer is involved in the escape of *A. fumigatus*-conidia from the innate immune cells recognition (52). However, since NK cells are not activated by *ΔrodA* (7), the mechanism underlying the recognition of *A. fumigatus* by immune cells might vary among immune cells.

Furthermore, we examined whether the antifungal activity elicited by Vγ9Vδ2 T cells was dependent on soluble factors secreted by Vγ9Vδ2 T cells. Vγ9Vδ2 T cells and NK cells are innate immune effector cells, and NK cells are known to exhibit antifungal activity against *A. fumigatus*, possibly through the activity of IFN-γ (16). In a previous study, Vγ9Vδ2 T cells secreted a high level of TNF-α in response to water extracts of *A. fumigatus* (21), and Th1 immunity is considered important for *Aspergillus* infections (10). Therefore, we examined the level of IFN-γ and TNF-α secreted from Vγ9Vδ2 T cells in response to *A. fumigatus*-hyphae. Surprisingly, *A. fumigatus*-hyphae failed to induce the secretion of these cytokines from Vγ9Vδ2 T cells but instead suppressed these cytokine levels when Vγ9Vδ2 T cells were stimulated by HMBPP. In addition, supernatants from the cell culture of *A. fumigatus*-hyphae with Vγ9Vδ2T cells had little antifungal activity, strongly indicating that the antifungal activity of Vγ9Vδ2 T cells was not mediated by soluble factors, such as IFN-γ or TNF-α.

We further examined whether the antifungal activity of Vγ9Vδ2 T cells is dependent on direct contact with *A. fumigatus*-hyphae. Based on the results of the standard cell culture insert system, which allows for only small soluble molecules to pass through the membrane, the Vγ9Vδ2 T cells exhibited the antifungal activity through direct contact with *A. fumigatus*-hyphae. Subsequently, we confirmed the degranulation of Vγ9Vδ2 T cells by monitoring the mobility of CD107a upon contact with *A. fumigatus*-hyphae (33). In addition, the inhibition of degranulation with EGTA significantly decreased the antifungal effect exhibited by Vγ9Vδ2 T cells, indicating the involvement of degranulation in the antifungal activity of Vγ9Vδ2 T cells. CD107a-positive intracellular vesicles contain at least three types of effector molecules: granzymes, granulysin, and perforins (53–55). These mediate cytotoxicity against bacteria and tumor cells through differential mechanisms at the immunological synapse (56, 57). Granulysin has been indicated to exert antifungal activity against *Cryptococcus neoformans* (58, 59). In addition, the antifungal activity of NK cells against *A. fumigatus*-hyphae is induced by perforin (17).

This study has some limitations. First, we could not prove the specific type of effector molecules in CD107a-positive intracellular vesicles critical for antifungal activity mediated by Vγ9Vδ2 T cells. Second, although it was indicated that GM itself was not recognized by Vγ9Vδ2 T cells, the antigenic entity of *A. fumigatus* and *Mucorales* such as *C. bertholletiae* and *R. microsporus* for Vγ9Vδ2 T cells remains unidentified. Third, we prepared hyphal layers resistant to antifungal agents on a polystyrene plate by culturing conidial suspensions as previously reported (42, 43). We referred to these as "biofilms" in the Discussion. In this study, while we confirmed hyphal layers, composed of many mycelia, under an optical microscope, we did not validate the presence of extracellular matrix or perform high-resolution microscopic evaluation.

In conclusion, this study strongly indicates that Vγ9Vδ2 T cell-based adoptive cell therapy could be a novel candidate for the next-generation therapeutics for IA. In addition, immunotherapy using Vγ9Vδ2 T cells may also be effective against other filamentous fungal infections. Given that many individuals with IA are highly immunocompromised, developing immunotherapy that potentially bolsters damaged or exhausted immune systems should be developed for the treatment of IA patients. We

are currently planning to establish *in vivo* experimental systems to examine the effects of Vγ9Vδ2 T cells on IA.

## MATERIALS AND METHODS

### Derivation of Vγ9Vδ2T cells and NK cells

Human PBMCs were obtained from healthy adult volunteers. Vγ9Vδ2 T cells were derived by stimulating PBMCs with PTA and IL-2 for 11 days. NK cells were *ex vivo* expanded by treating CD3-depleted PBMCs with IL-2/IL-18 for 10 days. Detailed methods for Vγ9Vδ2 T cells and NK cells expansion are described in Supplementary Methods. All experiments were conducted using Vγ9Vδ2 T cells derived from at least two different donors, demonstrating the generality of the mechanism underlying the antifungal activity of Vγ9Vδ2 T cells.

### Preparation of fungal strains

A previously characterized *A. fumigatus* strain (*Af293*) was used in this study. All filamentous fungi were grown on potato-dextrose-agar (PDA) slants for 4–7 days at 37°C. Subsequently, spores were harvested by gently scraping the surface of the slants and re-suspended in PBS supplemented with 0.05% Tween 20. Hyphae were removed by passing the suspension through a cell strainer (0.4 mm nylon mesh pore membrane). The conidial suspension was washed twice and re-suspended in RPMI1640 medium. The fungal suspension was used as a target for Vγ9Vδ2 T cells. Suspensions of filamentous fungi species-derived conidia were adjusted to a concentration of $1 \times 10^6$ /mL, which were dispensed into wells at a volume of 300 µL per well. Growth of hyphae was achieved by cultivating the conidial suspension at 37°C for 24 h in polystyrene culture plates. However, *A. terreus* was incubated for 96 h owing to its low growth speed. In addition to *Af293*, we used *Δuge5*, an *A. fumigatus* mutant, deficient in the *uge5* gene, resulting in the absence of Gal*f* necessary for GM synthesis and, by extension, the absence of GM (60). Additionally, we used several azole-resistant *A. fumigatus* strains, each with a different resistance mechanism: *NGS-ER7*, *MF2108*, and *IFM63240*. *NGS-ER7* is an environmental strain isolated in Europe. It shows azole resistance potentially attributed to a tandem repeat in the promoter region of *cyp51A* (TR46), *cyp51A* gene mutations (Y121F and T289), and *hmg1* gene mutation (E105K, S212P, and Y564H) [minimum inhibitory concentration (MIC): itraconazole = 2, voriconazole >8] (61). In contrast, *MF2108* and *IFM63240* are clinically isolated strains whose azole resistance is potentially owing to mutations other than *cyp51A* mutation. *MF2108* (MIC: itraconazole >8, voriconazole being >8) has one splice site mutation in the *HapE* gene (62), whereas *IFM63240* (MIC: itraconazole >8, voriconazole = 4) has two mutation (S269F in *hmg1* gene and A350T in *erg6* gene) (63). Besides *A. fumigatus*, we included other filamentous fungi causing invasive infections, such as *A. terreus* (*ATCC10690*), *A. niger* (*ATCC6275*), *A. flavus* (*ATCC9643*), *C. bertholletiae* (*MF2710*), *R. oryzae (MF1166), R. microsporus* (*IFM46417*), *M. circinelloides* (*IFM55051*), and *R. pusillus* (*IFM40783*). *Δuge5* was kindly donated by Dr. Donald C. Sheppard and *IFM63240, IFM46417, IFM55051*, and *IFM40783* by Chiba University. *MF2710* and *MF1166* were clinical strains collected in Nagasaki University Hospital.

### Assessment of antifungal activity

Antifungal activity against filamentous fungal hyphae was assessed through colorimetric assay with XTT (Merck & Co., Darmstadt, Hesse, Germany), in which the absorbance correlated with the number of viable fungal cells. First, Vγ9Vδ2 T cells were added to *A. fumigatus* or other filamentous fungi hyphae suspension at various effector-to-target ratios and incubated for 24 h. Wells were washed with PBS, and distilled water was added to lyse Vγ9Vδ2 T cells. After the suspension was placed at room temperature for 30 min, an XTT labeling mixture was added to each well at a final concentration of 0.3 mg/mL and incubated at 37°C with 5% $CO_2$. After incubation for 24 h, 100 µL of supernatant

samples was transferred to a 96-well flat-bottom plate in duplicate, and the optical density (OD) was measured at 450 nm with a 655-nm reference filter. The antifungal effect was calculated using the following formula: the percent damage = $(1 - X/Y) \times 100$, where $X$ is the OD of the samples under the various culture conditions, and $Y$ is that of the negative control (fungus alone). In addition, the antifungal activity exhibited by Vγ9Vδ2 T cells before cryopreservation was evaluated under the same protocol. After the co-culture of filamentous fungi and Vγ9Vδ2 T cells at different durations under the aforementioned conditions, the plate was centrifuged at 5,400 rpm for 2 min, and the supernatants were stored at $-80°C$.

Cell culture inserts (Nunc) were used to evaluate the effects of the direct contact between Vγ9Vδ2 T cells and fungi and those of soluble factors released from Vγ9Vδ2 T cells on antifungal activity. Given that the cell culture inserts possess pores allowing soluble factors to pass through but preventing Vγ9Vδ2 T cells and fungi from passing, the following culture combinations were employed: Vγ9Vδ2 T cells + fungus//fungus (without contact), fungus//Vγ9Vδ2T cells + fungus (with contact), and fungus//fungus (negative control). The XTT assay evaluated the fungal damage in the right compartment.

To evaluate the effect of HMBPP, which potently induces the release of Th1-type cytokines from Vγ9Vδ2 T cells (28), on the antifungal activity of Vγ9Vδ2 T cells, serially diluted HMBPP solutions were added to the wells when the Vγ9Vδ2 T cells were added to the fungal solution. To examine the effect of exocytosis of cytotoxic lysosomal proteins in fungal damage, Vγ9Vδ2 T cells were preincubated with 0.5–10 mM EGTA (Merck & Co.), a $Ca^{2+}$ chelating agent, for 30 min at room temperature before the antifungal assay, leading to the inhibition of $Ca^{2+}$ flux required for degranulation (36, 37).

To evaluate the antifungal activity of Vγ9Vδ2 T cells against *Aspergillus* conidia, resting conidia were incubated with Vγ9Vδ2 T cells for 6 h. Subsequently, 100 µL of suspensions was spread over PDA agar plates and incubated at 37°C. After 24 h, the number of colonies was counted. Conidial damage was assessed based on the CFUs per mL.

## Cytokine assay

IFN-γ and TNF-α levels in the culture supernatants of fungi and Vγ9Vδ2 T cells co-incubated for 24 h were measured using an enzyme-linked immunosorbent assay kit (Pepro-Tech, Cranbury, NJ) according to the manufacturer's instructions. In brief, capture mAb for human IFN-γ or TNF-α (1 µg/mL in PBS) were placed into the wells of a 96-well flat-bottom plastic plate (Thermo Fisher Scientific, Waltham, Massachusetts, USA). After the plate was incubated at ambient temperature overnight, the wells were washed four times with 300 µL of PBS/0.05% Tween 20 (Nacalai Tesque Inc.) and 300 µL of PBS/1% bovine serum albumin (BSA fraction V, Nacalai Tesque Inc.) was added to the wells. The plate was incubated at room temperature for 2 h, and the wells were washed four times with 300 µL of PBS/0.05% Tween 20. Furthermore, 100 µL of culture supernatants and a serially diluted human IFN-γ or TNF-α standard solution were added to the wells and incubated at room temperature for 2 h. Subsequently, the wells were washed four times with 300 µL of PBS/0.05% Tween 20. Biotin-conjugated detection mAb for human IFN-γ or TNF-α was diluted with PBS/0.05% Tween 20/0.1% BSA for a final concentration of 1 µg/mL, and 100 µL of this solution was added to the wells. After incubation at room temperature for 2 h, the wells were washed four times with 300 µL of PBS/0.05% Tween 20. Furthermore, 100 µL of horseradish peroxidase-conjugated avidin solution was added to the wells and incubated at room temperature for 30 min. The wells were washed four times with 300 µL of PBS/0.05% Tween 20, 100 µL of 2,2'-azino-bis (3-ethylbenzothiazoline-6-sulfonic acid) diammonium salt liquid substrate (Merck & Co.) was added, and the plate was incubated at room temperature for 5 min. Finally, 100 µL of 1% sodium dodecyl sulfate solution in water was added to the wells, and the OD at 405 nm was measured using a NIVO multiplate reader (Revvity, Yokohama, Kanagawa, Japan). As a positive control, HMBPP was added to the mixture of Vγ9Vδ2 T cells and fungi at a final concentration of 100 nM.

## Flowcytometric analysis of CD107a on Vγ9Vδ2 T cells

To evaluate the degree of degranulation in Vγ9Vδ2 T cells, the mobilization of CD107a was assessed using a flow cytometer, as this membrane protein translocates to the cell surface during the degranulation process (31). A serially diluted *Af293* suspension (25 µL) was placed into a 96-well round-bottom plastic plate (Corning Inc., Corning, NY). After incubating at 37℃ for 24 h for hyphae formation, 25 µL of Vγ9Vδ2 T cell suspensions was added to the wells at different effector-to-target ratios, followed by the addition of 5 µL of phycoerythrin-conjugated mouse anti-human CD107a mAb (BioLegend, San Diego, CA).

As controls, supernatants from co-culture of Vγ9Vδ2 T cells with *A. fumigatus*-hyphae for 24 h were also included. The plate was briefly centrifuged at 500 rpm for 2 min and incubated at 37℃ for 2 h. Subsequently, 3 µL of FITC-conjugated mouse anti-human TCR Vδ2 mAb was added to the wells and incubated for an additional 15 min. Finally, Vγ9Vδ2 T cells were harvested and analyzed through a FACSLyric flow cytometer (Becton Dickinson & Co.). The cell population was visualized using a FlowJo software ver. 10 (FlowJo LLC).

## Statistical analysis

Statistical significance was determined using the unpaired, two-sided Student *t*-test, implemented in a GraphPad Prism software ver. 8.4.3. *P*-value < 0.05 was considered statistically significant. Bars show arithmetic means of the values of the independent experiment ± standard error of the mean.

## ACKNOWLEDGMENTS

We thank Ms. Ishida for her assistance in the experiments. The funders had no role in study design, data collection and interpretation, or the decision to submit the work for publication.

This research was supported by AMED, Grant No. 22fk0108530h0001 to T.T., Grant No. A48 and A90 to Y.T, by JSPS KAKENHI Grant-in-Aid for Scientific Research (C), Grant No. 23K07941 to T.T., by JST START University Ecosystem Promotion Type (Supporting the Creation of Startup Ecosystems in Startup Cities), Japan, Grant No. JPMJST2281 to Y.T., by Grants-in-Aid for Scientific Research from MEXT, Grant No. 16K08844 and 23K06677 to Y.T., and by Research Funding granted by Nagasaki University President Shigeru Kohno to Y.T.

S.K., Conceptualization, Investigation, Validation, Writing—original draft | T.K., Conceptualization, Supervision, Validation, Formal analysis, Data curation, Writing—review and editing, Funding acquisition, project administration | H.N., Formal analysis | D.O., Formal analysis | Y.I., Validation | N.N., Validation | T.H., Formal analysis | K. Takeda, Formal analysis | S.I., Formal analysis | N.I., Validation | M.T., Formal Analysis | N.S., Formal analysis, Validation | A.W., Writing—review and editing | K.I., Writing—review and editing | K.Y., Writing—review and editing | Y.T Tanaka, Supervision, Formal analysis, Data curation, Project administration, Writing—review and editing | H.M., Writing—review and editing, Project administration.

## AUTHOR AFFILIATIONS

[1]Department of Respiratory Medicine, Graduate School of Biomedical Sciences, Nagasaki University, Nagasaki, Japan
[2]Department of Respiratory Medicine, Nagasaki University Hospital, Nagasaki, Japan
[3]Department of Infectious Diseases, Nagasaki University Graduate School of Biomedical Sciences, Nagasaki, Japan
[4]Health Center, Nagasaki University, Nagasaki, Japan
[5]Department of Pharmacotherapeutics, Nagasaki University Graduate School of Biomedical Sciences, Nagasaki, Japan

⁶Infectious Diseases Experts Training Center, Nagasaki University Hospital, Nagasaki, Japan
⁷Medical Mycology Research Center, Chiba University, Chiba, Japan
⁸Department of Laboratory Medicine, Nagasaki University Hospital, Nagasaki, Japan
⁹Center for Medical Innovation, Nagasaki University, Nagasaki, Japan

## AUTHOR ORCIDs

Satoru Koga  http://orcid.org/0009-0006-9415-2305
Takahiro Takazono  http://orcid.org/0000-0002-0696-5386
Tatsuro Hirayama  http://orcid.org/0000-0001-8554-3482
Masato Tashiro  https://orcid.org/0000-0001-7609-7679
Akira Watanabe  http://orcid.org/0000-0002-3057-2937

## FUNDING

| Funder | Grant(s) | Author(s) |
|---|---|---|
| Japan Agency for Medical Research and Development (AMED) | 22fk0108530h0001 | Takahiro Takazono |
| MEXT \| Japan Society for the Promotion of Science (JSPS) | 23K07941 | Takahiro Takazono |
| MEXT \| Japan Society for the Promotion of Science (JSPS) | A48 | Yoshimasa Tanaka |
| MEXT \| Japan Society for the Promotion of Science (JSPS) | A90 | Yoshimasa Tanaka |
| Ministry of Education, Culture, Sports, Science and Technology (MEXT) | JPMJST2281 | Yoshimasa Tanaka |
| Nagasaki University | 16K08844 | Yoshimasa Tanaka |
| Nagasaki University | 23K06677 | Yoshimasa Tanaka |

## AUTHOR CONTRIBUTIONS

Satoru Koga, Conceptualization, Investigation, Validation, Writing – original draft | Takahiro Takazono, Conceptualization, Data curation, Formal analysis, Funding acquisition, Project administration, Supervision, Validation, Writing – original draft | Hodaka Namie, Formal analysis | Daisuke Okuno, Formal analysis | Yuya Ito, Validation | Nana Nakada, Validation | Tatsuro Hirayama, Formal analysis | Kazuaki Takeda, Formal analysis | Shotaro Ide, Formal analysis | Naoki Iwanaga, Validation | Masato Tashiro, Formal analysis | Noriho Sakamoto, Formal analysis | Akira Watanabe, Writing – review and editing | Koichi Izumikawa, Writing – review and editing | Katsunori Yanagihara, Writing – review and editing | Yoshimasa Tanaka, Data curation, Formal analysis, Project administration, Supervision, Writing – review and editing | Hiroshi Mukae, Project administration, Writing – review and editing

## ADDITIONAL FILES

The following material is available online.

### Supplemental Material

**Supplemental Fig. S1 to S4 (Spectrum03614-23-S0001.pdf).** Profiles of Vγ9Vδ2 T cells.
**Supplemental methods (Spectrum03614-23-S0002.docx).** Derivation of Vγ9Vδ2T cells and NK cells.

Open Peer Review

**PEER REVIEW HISTORY (review-history.pdf).** An accounting of the reviewer comments and feedback.

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
