## [Reviewer comments · Microbiology Spectrum]

Microbiology Spectrum

Human V γ 9V δ 2 T cells exhibit antifungal activity against *Aspergillus fumigatus* and other filamentous fungi

Satoru Koga, Takahiro Takazono, Hodaka Namie, Daisuke Okuno, Yuya Ito, Nana Nakada, Tatsuro Hirayama, Kazuaki Takeda, Shotaro Ide, Naoki Iwanaga, Masato Tashiro, Noriho Sakamoto, Akira Watanabe, KOICHI IZUMIKAWA, Katsunori Yanagihara, Yoshimasa Tanaka, and Mukae Hiroshi

Corresponding Author(s): Takahiro Takazono, Nagasaki Daigaku Daigakuin Ishiyakugaku Sogo Kenkyuka

Review Timeline:

Submission Date:	October 9, 2023
Editorial Decision:	November 12, 2023
Revision Received:	January 3, 2024
Editorial Decision:	January 28, 2024
Revision Received:	January 29, 2024
Accepted:	February 11, 2024

Editor: Paschalis Vergidis

Reviewer(s): Disclosure of reviewer identity is with reference to reviewer comments included in decision letter(s). The following individuals involved in review of your submission have agreed to reveal their identity: Sebastian Thomas Wurster (Reviewer #1)

Transaction Report:

DOI: <https://doi.org/10.1128/spectrum.03614-23>

Re: Spectrum03614-23 (Human V γ 9V δ 2 T cells exhibit antifungal activity against *Aspergillus fumigatus* and other filamentous fungi)

Dear Dr. Takahiro Takazono:

Thank you for the privilege of reviewing your work. Below you will find my comments, instructions from the Spectrum editorial office, and the reviewer comments.

Revision Guidelines

Sincerely,
Paschalis Vergidis
Editor
Microbiology Spectrum

Reviewer #1 (Comments for the Author):

Koga et al. studied the antifungal activity of $\gamma\delta$ T cells against *Aspergillus fumigatus* and other filamentous fungi. This topic is unique and interesting, given that (i) the contribution of these cells to natural antifungal immunity is poorly studied and (ii) ex-vivo expanded $\gamma\delta$ T cells could be used for antifungal immunotherapy in the future. However, there are several conceptual and experimental issues as well as issues related to data presentation and over-interpretation that need to be addressed.

The manuscript clearly suggests that the authors' focus was the use of $\gamma\delta$ T cells as an immunotherapy, which also aligns with the use of an expanded $\gamma\delta$ T-cell formulation for their experiments. Nonetheless, the authors need to make a clearer point in the Discussion that the properties of primary, unexpanded $\gamma\delta$ T cells in vivo might be vastly different. Along these lines, given that the authors had obtained > 1 million $\gamma\delta$ T cells at baseline (before expansion), it would have been interesting to perform at least some selected experiments with unexpanded cells to gain insights into the natural role of these cells in antifungal immunity. In particular, it would have been beneficial to compare the expression of various PRRs and pro-apoptotic effectors (lines 126-134), and possibly also exhaustion markers (which might be induced during ex-vivo expansion) in unexpanded and expanded cells.

Given the focus on an "off-the-shelf" cell therapy, the use of cryopreserved cells for the authors' experiments makes sense. Nonetheless, cryopreservation might be another confounder and might alter cell functionality, which poses further limitations on the applicability of the findings to the actual in-vivo function of unexpanded cells. This should be discussed.

Overall, I think the issue of antifungal/azole-resistant strains and activity of $\gamma\delta$ T cells against such strains is overstated. The common mutations facilitating such resistance are unrelated to cell wall biochemistry and architecture; therefore, the baseline expectation would be that host recognition of cell wall structures (e.g., GAG or beta-glucans) remains unaffected by the isolates' azole resistance.

Some assays use very high E/T ratios. While it is understood that factors like plate geometry might necessitate such E/T ratios, it would have been beneficial to include co-culture studies with a known fungicidal cell type (e.g., NK cells or even crude PBMCs) as a control to put the fungicidal activity of $\gamma\delta$ T cells into perspective. Although the overall immunotherapeutic impact of such cells would likely extend beyond direct fungal killing (e.g., via cytokine/chemokine release to attract other effector cells), the authors place a strong focus on fungal killing/fungal growth inhibition. Therefore, such an additional control would be essential and would have been feasible to include (even from the same donor), given the use of PBMCs as a starting point. Moreover, some of the figures (e.g., 1A and 1B) do not even indicate the E/T ratio, which needs to be addressed.

How did the authors induce or confirm the presence of "biofilms" in Fig. 2? What was the difference in the experimental design between Fig. 1 and Fig. 2? Technically, although *A. fumigatus* can form biofilms, a hyphal layer at the bottom of the plate (without any additional components or extracellular matrix) might not constitute a biofilm.

I cannot follow the authors' conclusions drawn from Fig. 3. While the use of different assays is reasonable due to planktonic growth of Δ uge3, the impact of $\gamma\delta$ T cells on both Δ uge3 and Δ uge5 was minimal and non-significant. Therefore, the authors cannot reasonably conclude that GAG but not GM is involved in recognition by $\gamma\delta$ T cells. Moreover, the experiment shown does not suffice to conclusively demonstrate that $\gamma\delta$ T cells recognize GAG, given that the mutant has significantly altered growth patterns and substantially altered biochemical and structural properties. To make such an argument, the authors would have needed to stimulate the cells with purified fungal cell wall components and measured surrogates of activation (e.g., cytokine release) as a 2nd line of evidence. In addition, Fig. 1B suggests activity of $\gamma\delta$ T cells against some Mucorales (e.g., *Cunninghamella*), although Mucorales do not produce GAG. Therefore, it is unlikely that GAG is the sole or primary driver of the $\gamma\delta$ T-cell response. I suggest deleting this part or at least amending and substantially toning down the conclusions and adding more nuanced discussion.

I do not like the term "Th1 cytokines" for readouts shown in Fig. 4, i.e., an experiment not using conventional Th cells. IFN- γ is secreted by various cell types and is not solely a Th1 cytokine. The same applies to similar references to Th1 cytokines in the Discussion, e.g., in lines 277 and 340.

The experiment shown in Fig. 4 should have been analyzed with a multiplex panel or at least several ELISAs. For instance, why did the authors only cite that TNF- α may be produced but not actually confirm this in their experiment?

The sentence in Line 207ff. (Notably,...) should be deleted or re-written, as the corresponding experiment (Fig. 4B) did not assess fungal killing.

All figures except Fig. 3 are purely descriptive and lack indicators of statistical significance. Significance testing has been described but not added to the panels.

Fig. 5 would need panels (e.g., columns + error bars) that show the effect of both direct co-culture and supernatants on CD107a expression (in addition to the flow panels from a representative dataset). It is certainly not ideal to show such a panel for a technical/mechanistic control (Fig. 5B) while not showing the actual biological effect and its reproducibility.

The authors need to provide a clear statement (in Materials & Methods and each figure legend) how many independent biological replicates (i.e., cells from different donors) and technical replicates were performed.

Throughout the paper, the authors need to refer to specific panels (e.g., Fig. 4A) and not only to figures as a whole (e.g., Fig. 4) when presenting or discussing specific results.

Minor Comments:

The Introduction is somewhat unfocused. First, the long first paragraph goes back and forth between *Aspergillus* and Mucorales. Second, given that this is a predominantly immunologic/mechanistic paper, there is no need to spend 21 lines on the ineffectiveness of antifungal therapies and need for adjunct immunotherapies. This point can be made much more concisely, as emerging resistance and limitations of antifungal therapy are well known to the readership of *Microbiology Spectrum*. Third, the rationale for antifungal immunotherapy is not only based on the limitations of antifungal therapy but on the fact that outcomes of invasive mold infections in severely immunocompromised patients are predominantly host driven (i.e., remission of underlying hematological malignancies and recovery of cytopenia). There is abundant literature on this phenomenon, especially from U.S. cancer centers. Forth, the properties and known functions of $\gamma\delta$ T cells, especially in the context of infectious diseases, should be introduced more comprehensively. To that end, I would suggest putting the content of lines 266-284 in the Introduction.

Several statements in the Discussion, including references to the authors' own prior work, require references, e.g., in lines 266-273 or lines 285-295.

The Materials & Methods section is overly detailed, especially with regard to standard methodology (e.g., isolation of PBMCs). These sections might be shortened or relegated to a Supplementary Methods section.

The manuscript is overall well-written but some editing would be needed to correct minor grammar mistakes and stylistic issues.

January 3, 2024

Dear Editor,

The authors thank the Reviewers for their valuable comments. We have now revised our manuscript to address these comments with point-by-point responses.

Reviewer 2's comment 1: The manuscript clearly suggests that the authors' focus was the use of $\gamma\delta$ T cells as an immunotherapy, which also aligns with the use of an expanded $\gamma\delta$ T-cell formulation for their experiments. Nonetheless, the authors need to make a clearer point in the Discussion that the properties of primary, unexpanded $\gamma\delta$ T cells in vivo might be vastly different. Along these lines, given that the authors had obtained > 1 million $\gamma\delta$ T cells at baseline (before expansion), it would have been interesting to perform at least some selected experiments with unexpanded cells to gain insights into the natural role of these cells in antifungal immunity. In particular, it would have been beneficial to compare the expression of various PRRs and pro-apoptotic effectors (lines 126-134), and possibly also exhaustion markers (which might be induced during ex-vivo expansion) in unexpanded and expanded cells.

Authors' responses: As Reviewer 2 pointed out, it is important to gain insight into the physiological role of $\gamma\delta$ T cells in a stationary phase by comparing various markers before and after expansion. Therefore, we examined several cell surface markers, including pattern recognition receptors (PPRs) (Dectin-1, TLR-2, and TLR-4), natural killer cell markers (NKG2D, DNAM-1, FasL, and TRAIL), CD25, an activation marker for expanded culture of V γ 9V δ 2 T cells, and a T cell exhaustion maker (PD-1), on $\gamma\delta$ T cells on day 0, day 3, and day 11 after stimulation of PBMC with PTA/IL-2. The aforementioned PRRs were selected because of their association with *Aspergillus spp.* One was Dectin-1, one of the most well characterized PRRs, was chosen, and the others (TLR2 and TLR4) were selected based on previous reports indicating that $\gamma\delta$ T cells express the TLRs. The staining profiles were included in Supplementary Fig. 4, and the following sentences were added to the text.

Line 143-153 in Results: In addition, we examined several cell surface makers expressed on V γ 9V δ 2 T cells derived from peripheral blood stimulated with PTA/IL-2 for 0, 3, and 11 days to gain insight into the possible functions of V γ 9V δ 2 T cells in stationary and activated phases. As shown in Supplementary Fig. 4, the expression of CD369 (Dectin-1, CLEC7A), CD282 (TLR-2), CD284 (TLR-4), FasL, and TRAIL were low at all the time points. In contrast, a high level of DNAM-1 expression was observed at all time points. The expression of NKG2D was initially high but decreased on day 3 after stimulation and subsequently resumed to an initial

level on day 11. Conversely, CD25 was not expressed in the steady state, but was expressed to a markedly high level on day 3 after stimulation, and the expression gradually decreased thereafter. Programmed death-1 (PD-1, CD279) was slightly expressed on day 0, and the expression level increased on day 3 and returned to the stationary state on day 11.

Line 322-332 in Discussion: In addition, we examined V γ 9V δ 2 T cells for the expression of several cell surface markers, including PRRs (Dectin-1, TLR2 and TLR4), which have been reported to be associated with *Aspergillus* infections (50), NK cell-related markers, NKG2D and DNAM-1, cell death receptor ligands (FasL and TRAIL), CD25, an activation marker of V γ 9V δ 2 T cells, and PD-1, a T cell exhaustion marker, on days 0, 3, and 11 after stimulation with PTA/IL-2. Dectin-1 is one of the most well-characterized PRRs recognizing b-D-glucans (50, 51), whereas its expression on V γ 9V δ 2 T cells had not been reported. It was demonstrated that TLR2 and TLR4 were upregulated on V γ 9V δ 2 T cells after stimulation with TLR4 ligands (52), leading to induced antibacterial responses (53). In this study, PRRs were not expressed on V γ 9V δ 2 T cells under a condition we employed. In addition, the expression levels of other cell surface markers remained noticeably unchanged before and after the expanded culture, except for NKG2D, CD25 and PD-1.

Reviewer 2's comment 2: Given the focus on an "off-the-shelf" cell therapy, the use of cryopreserved cells for the authors' experiments makes sense. Nonetheless, cryopreservation might be another confounder and might alter cell functionality, which poses further limitations on the applicability of the findings to the actual in-vivo function of unexpanded cells. This should be discussed.

Authors' responses: As pointed out by Reviewer 2, we compared the antifungal activity of freshly prepared and cryopreserved $\gamma\delta$ T cells to examine the effect of cryopreservation. Although this finding was not included in the original manuscript, we have added it to Figure 1A (the rightmost bar). Additionally, the following sentences have been incorporated into the revised manuscript.

Line 159-160 in Results: and antifungal activity of V γ 9V δ 2 T cells after cryopreservation was comparable to that before cryopreservation.

Line 435-436 in Materials and Methods: In addition, the antifungal activity exhibited by V γ 9V δ 2 T cells before cryopreservation was evaluated under the same protocol.

Line 753-754 in the legend of Fig. 1A: and cryopreservation did not significantly alter the antifungal activity of V γ 9V δ 2 T cells.

Reviewer 2's comment 3: Overall, I think the issue of antifungal/azole-resistant strains and activity of $\gamma\delta$ T cells against such strains is overstated. The common mutations facilitating such resistance are unrelated to cell wall biochemistry and architecture; therefore, the baseline expectation would be that host recognition of cell wall structures (e.g., GAG or beta-glucans) remains unaffected by the isolates' azole resistance.

Authors' responses: We agree with Reviewer 2 regarding the statement on relationship between drug resistance and cell-wall constitution. In the original manuscript, we aimed to highlight the effect of $\gamma\delta$ T cells against azole-resistant *Aspergillus fumigatus*, which is resistant to the treatment with conventional antifungal agents, and to demonstrate that $\gamma\delta$ T cells exhibit antifungal activities distinctively from conventional antifungal agents. However, as pointed out by Reviewer 2, we overstated the relationship between drug-resistant strains and antifungal activity of $\gamma\delta$ T cells. Therefore, we have eliminated or shortened the statement of azole-resistant *A. fumigatus* as follows:

Lines 310-311 in Discussion: The existence of azole-resistant *A. fumigatus* has been reported worldwide (46), posing a clinically significant problem in future (7, 47).

And we added the following statement in the revised manuscript.

Lines 305-308 in Discussion: Since common mutations in *A. fumigatus* responsible for drug resistance are generally unrelated to the structure and components of cell walls that are highly antigenic for the host immune system, the antifungal activity of V γ 9V δ 2 T cells might have nothing to do with such azole-resistant mechanisms.

Reviewer 2's comment 4: Some assays use very high E/T ratios. While it is understood that factors like plate geometry might necessitate such E/T ratios, it would have been beneficial to include co-culture studies with a known fungicidal cell type (e.g., NK cells or even crude PBMCs) as a control to put the fungicidal activity of $\gamma\delta$ T cells into perspective. Although the overall immunotherapeutic impact of such cells would likely extend beyond direct fungal killing (e.g., via cytokine/chemokine release to attract other effector cells), the authors place a strong focus on fungal killing/fungal growth inhibition. Therefore, such an additional control would be essential and would have been feasible to include (even from the same donor), give the use of PBMCs as a starting point.

Moreover, some of the figures (e.g., 1A and 1B) do not even indicate the E/T ratio, which needs to be addressed.

Authors' responses: As Reviewer 2 pointed out, controls are important to justify the conditions we used. We evaluated the antifungal activity of NK cells which have been reported to exhibit potent antifungal activity against *Aspergillus fumigatus* in vitro using the same method (XTT assay). The results have been added to Fig. 1A. In addition, the derivation of NK cells was briefly described in Materials and Methods, while more detailed methods were included in Supplementary Methods, alongside those for V γ 9V δ 2 T cells. The revised sentences are as follows:

Lines 154-158 in Results: We subsequently examined the effect of V γ 9V δ 2 T cells on the hyphae of *A. fumigatus*. When *A. fumigatus* filamentous hyphae were cultured in the presence of V γ 9V δ 2 T cells, the viability of *A. fumigatus* was significantly reduced, compared to that in the absence of V γ 9V δ 2 T cells, strongly indicating that V γ 9V δ 2 T cells have potent antifungal activity, which was comparable to that of NK cells (Fig. 1A).

Lines 296-301 in Discussion: The antifungal activity of V γ 9V δ 2 T cells against *A. fumigatus* was comparable to that of NK cells. The advantage of V γ 9V δ 2 T cells over NK cells is that the expansion rate of V γ 9V δ 2 T cells is much greater than that of NK cells. When PBMCs are stimulated with PTA/IL-2, a large number of highly purified V γ 9V δ 2 T cells can be obtained within 11 days without purification, indicating that V γ 9V δ 2 T cell-based adoptive transfer therapy is more practical than that of NK cells.

Lines 386-390 in Materials and Methods: Derivation of V γ 9V δ 2 T cells and NK cells. Human PBMC were obtained from healthy adult volunteers. V γ 9V δ 2 T cells were derived by stimulating PBMC with PTA and IL-2 for 11 days. NK cells were ex-vivo expanded by treating CD3-depleted PBMC with IL-2/IL-18 for 10 days. Detailed methods for V γ 9V δ 2 T cells and NK cells expansion are described in Supplementary Methods.

Lines 751-753 in the legend of Fig. 1A: Dose-dependent antifungal activity of V γ 9V δ 2 T cells against Af293, a wild type *A. fumigatus* strain, which was comparable to that of NK cells.

As pointed out by Reviewer 2, the original manuscript lacked information on the concentration of fungi used in the experiments, and the E:T ratios were unspecified in some figures. The

amounts of fungi used in the experiments have been added to Materials and Methods as follows:

Lines 400-402 in Materials and Methods: Suspensions of filamentous fungi species-derived conidia were adjusted to a concentration of 1×10^6 /ml, which were dispensed into wells at a volume of 300 μ L per well.

Reviewer 2's comment 5: How did the authors induce or confirm the presence of "biofilms" in Fig. 2? What was the difference in the experimental design between Fig. 1 and Fig. 2? Technically, although *A. fumigatus* can form biofilms, a hyphal layer at the bottom of the plate (without any additional components or extracellular matrix) might not constitute a biofilm.

Authors' responses: As pointed out by Reviewer 2, we did not prove the presence of biofilms because we did not ascertain the existence of extracellular substrates nor perform high-resolution microscopic analysis. Our experiments were conducted under a condition described in a previous report, demonstrating that biofilms characterized by a multicellular structure with many mycelia intertwined were formed by incubating conidial suspensions on a polystyrene plate (Ref. 33). Our study observed that the "biofilms" were resistant to antifungal drugs, while the formation of the extracellular matrix remains unconfirmed. Therefore, we refrained from using the term "biofilm" as much as possible. We revised the manuscript as follows.

Lines 172-183 in Results:

Notably, *Aspergillus* hyphae formed hyphal layers with numerous mycelia intertwined on a polystyrene plate, and the mycelia within the structures were more resistant to antifungal agents including voriconazole than planktonic ones (33, 34). In this study, complex intertwining hyphal layers were observed under an optical microscope after conidial suspensions were incubated for 24 h on a polystyrene plate. We subsequently compared the antifungal effect of V γ 9V δ 2 T cells and voriconazole against *A. fumigatus* forming the hyphal layers. V γ 9V δ 2 T cells exerted antifungal activity against *A. fumigatus*-hyphae even in the presence of the hyphal layers. Alternatively, all species of *A. fumigatus*, including azole-susceptible and several azole-resistant strains, were resistant to voriconazole (Fig. 2). Based on these findings, V γ 9V δ 2 T cells could potentially be developed as a novel antifungal modality for the treatment of azole-resistant *A. fumigatus*, and wild-type strains. In addition, V γ 9V δ 2 T cells are effective against *A. fumigatus* forming hyphal layers resistant to antifungal agents.

Lines 308-310 in Discussion:

In addition, V γ 9V δ 2 T cells were effective against *A. fumigatus* forming complex intertwining hyphal layers resistant to antifungal agents.

Lines 371-375 in Discussion (limitation part):

Third, we prepared hyphal layers resistant to antifungal agents on a polystyrene plate by culturing conidial suspensions as previously reported (33, 34). We referred to these as “biofilms” in the Discussion. In this study, while we confirmed hyphal layers, composed of many mycelia, under an optical microscope, we did not validate the presence of extracellular matrix or perform high-resolution microscopic evaluation.

Reviewer 2's comment 6: I cannot follow the authors' conclusions drawn from Fig. 3. While the use of different assays is reasonable due to planktonic growth of Δ uge3, the impact of $\gamma\delta$ T cells on both Δ uge3 and Δ uge5 was minimal and non-significant. Therefore, the authors cannot reasonably conclude that GAG but not GM is involved in recognition by $\gamma\delta$ T cells.

Moreover, the experiment shown does not suffice to conclusively demonstrate that $\gamma\delta$ T cells recognize GAG, given that the mutant has significantly altered growth patterns and substantially altered biochemical and structural properties. To make such an argument, the authors would have needed to stimulate the cells with purified fungal cell wall components and measured surrogates of activation (e.g., cytokine release) as a 2nd line of evidence.

Authors' responses: As pointed out by Reviewer 2, it was inappropriate to conclude that GAG itself was recognized by $\gamma\delta$ T cells. Therefore, data on Δ uge3 was removed from Fig. 3 and the manuscript. Fig. 3B might also be misleading, but the figure demonstrated that $\gamma\delta$ T cells exhibited similar antifungal activity to Δ uge5 as to *Af293*. Therefore, we retained the result and the description on Δ uge5. In addition, the sentence in the Limitation part was revised as follows:

Lines 369-371 in Discussions (limitation part):

Second, although it was indicated that GM itself was not recognized by V γ 9V δ 2 T cells, the antigenic entity of *A. fumigatus* and some species of *Mucorale* for V γ 9V δ 2 T cells remains unidentified.

Reviewer 2's comment 7: In addition, Fig. 1B suggests activity of $\gamma\delta$ T cells against some

Mucorales (e.g., Cunninghamella), although Mucorales do not produce GAG. Therefore, it is unlikely that GAG is the sole or primary driver of the $\gamma\delta$ T-cell response. I suggest deleting this part or at least amending and substantially toning down the conclusions and adding more nuanced discussion.

Authors' responses: As Reviewer 2 pointed out, *Mucorales* used in our experiments did not produce GAG. Indeed, this fact contradicts our discussions on *Auge3*. Therefore, data and sentences on *Auge3* were removed from Fig. 3.

Reviewer 2's comment 8: I do not like the term "Th1 cytokines" for readouts shown in Fig. 4, i.e., an experiment not using conventional Th cells. IFN- γ is secreted by various cell types and is not solely a Th1 cytokine. The same applies to similar references to Th1 cytokines in the Discussion, e.g., in lines 277 and 340.

Authors' responses: As noted by Reviewer 2, "Th1 cytokines" may not be appropriate in the context of our discussion. We removed the term "Th1 cytokines", and added individual cytokine names (Lines 201-222, 278 and 773-774, and a title of Fig. 4).

Reviewer 2's comment 9: The experiment shown in Fig. 4 should have been analyzed with a multiplex panel or at least several ELISAs. For instance, why did the authors only cite that TNF- α may be produced but not actually confirm this in their experiment?

Authors' responses: As noted in the manuscript, Th1-type immunity plays a pivotal role in protective immune responses against IA (Ref. 10), and V γ 9V δ 2 T cells could produce a significant level of TNF- α in response to water-soluble extracts of *A. fumigatus* (Ref. 21). In addition, NK cells are known to exhibit antifungal activity against *A. fumigatus*, possibly through IFN- γ -mediated signals (Ref. 16). Therefore, we examined the levels of IFN- γ in the supernatants. However, as you point out, TNF- α was not evaluated in our study despite citing previous reports on TNF- α . Therefore, we evaluated the levels of TNF- α in the supernatants using the same method as for IFN- γ . The reproducibility of the experiment was also confirmed, and the results were included in Figs. 4C and 4D. In addition, to ensure consistency, Figs. 4A and 4B for IFN- γ were relocated to compare the cytokines in the same donor-derived V γ 9V δ 2 T cells with those for TNF- α . Accordingly, the text was revised accordingly (Lines 201-222, 278, 349-354, 460-482)

Reviewer 2's comment 10: The sentence in Line 207ff. (Notably,...) should be deleted or re-written,

as the corresponding experiment (Fig. 4B) did not assess fungal killing.

Authors' responses: As pointed out by Reviewer 2, the results in Fig. 4B did not yield any conclusion for fungal killing. Therefore, we deleted the following sentence: Notably, V γ 9V δ 2 T cells could kill *A. fumigatus* even if IFN- γ secretion was hampered by *A. fumigatus*-hyphae.

Reviewer 2's comment 11: All figures except Fig. 3 are purely descriptive and lack indicators of statistical significance. Significance testing has been described but not added to the panels.

Authors' responses: As noted by Reviewer 2, indicators of significant differences were missing in many Figs. We have added indicators of significant differences in Figs 1A, 2, 4A-F, and 5B, C.

Reviewer 2's comment 12: Fig. 5 would need panels (e.g., columns + error bars) that show the effect of both direct co-culture and supernatants on CD107a expression (in addition to the flow panels from a representative dataset).

Authors' responses: As Reviewer 2 pointed out, Fig. 5 was the only result with a single data set and was inadequate for not proving reproducibility and providing indicators of statistical significance. Therefore, we added Fig. 5B, which shows the data set in triplicate, including indicators of statistical significance. In addition, "medium-only" was replaced by "supernatant-only" which was not shown in the original manuscript as a control for Figs. 5A, B. The manuscript was revised as follows:

Lines 490-491 in Materials and Methods:

As controls, supernatants from co-culture of V γ 9V δ 2 T cells with *A. fumigatus*-hyphae for 24 h were also included.

Lines 245-253 in Results:

Therefore, we examined the levels of CD107a expression on V γ 9V δ 2 T cells after incubation with *A. fumigatus*-hyphae or culture supernatants derived from the co-culture system of V γ 9V δ 2 T cells and *A. fumigatus*-hyphae. As shown in Figs. 5A and 5B, the expression of CD107a on V γ 9V δ 2 T cells was significantly upregulated when V γ 9V δ 2 T cells were co-cultured with *A. fumigatus*-hyphae, compared to those treated with supernatants of the co-culture. Notably, the upregulation of CD107a on V γ 9V δ 2 T cells was dependent on the dose of *A. fumigatus*-hyphae (Figs. 5A, 5B), confirming that the cell-to-cell contact between V γ 9V δ 2

T cells and *A. fumigatus*-hyphae is essential for the antifungal activity of V γ 9V δ 2 T cells.

Reviewer 2's comment 13: It is certainly not ideal to show such a panel for a technical/mechanistic control (Fig. 5B) while not showing the actual biological effect and its reproducibility.

Authors' responses: As pointed out by Reviewer 2, the mechanism underlying the EGTA pretreatment was unclear in the original manuscript. CD107a is expressed in intracellular vesicles that contain effector molecules, such as perforin and granzyme B. When degranulation occurs, the membranes of intracellular vesicles fuse with the outer cell membranes and CD107a in the intracellular membranes are mobilized onto the outer cell membranes, resulting in the expression of CD107a on the cell surface. Notably, the degranulation process depends on Ca²⁺ (Refs. 37, 38), and EGTA inhibits degranulation through chelating Ca²⁺ (Refs. 39, 40). This assay is used in many studies on the effector functions of immune cells, including $\gamma\delta$ T cells (Ref. 40). We performed EGTA assay to inhibit the degranulation of $\gamma\delta$ T cells based on previous studies. We revised the manuscript so that readers understand the reason why we conducted the EGTA pretreatment experiments as follows:

Lines 259-265 in Results: Ethylene glycol-bis(2-aminoethylether)-*N, N, N', N'*-tetra acetic acid A (EGTA) inhibits degranulation owing to its calcium-chelating effect (39, 40). To examine the effect of degranulation in the antifungal activity of V γ 9V δ 2 T cells, we evaluated the antifungal activity of V γ 9V δ 2 T cells after pretreatment with EGTA. As illustrated in Fig. 5C, the antifungal activity of V γ 9V δ 2 T cells against *A. fumigatus* was significantly suppressed by the pretreatment with EGTA, strongly indicating that degranulation was inexorably linked to the antifungal activity of V γ 9V δ 2 T cells.

Lines 359-361 in Discussion:

In addition, the inhibition of degranulation with EGTA significantly decreased the antifungal effect exhibited by V γ 9V δ 2 T cells, indicating the involvement of degranulation in the antifungal activity of V γ 9V δ 2 T cells.

In addition, we added the following sentences to provide a statement regarding biological and technical replicates.

Lines 390-392 in Materials and Methods:

All experiments were conducted using V γ 9V δ 2 T cells derived from at least two different donors, demonstrating the generality of the mechanism underlying the antifungal activity of

V γ 9V δ 2 T cells.

Lines 759, 764-765, 773, 793 and 804 in Figure legends:

Data shown are representative of at least three independent experiments.

Reviewer 2's comment 14: The authors need to provide a clear statement (in Materials & Methods and each figure legend) how many independent biological replicates (i.e., cells from different donors) and technical replicates were performed.

Authors' responses: As pointed out by Reviewer 2, we provided a statement regarding biological and technical replicates as follows:

Lines 390-392 in Materials and Methods:

All experiments were conducted using V γ 9V δ 2 T cells derived from at least two different donors, showing the generality of the mechanism underlying the antifungal activity of V γ 9V δ 2 T cells.

Lines 759, 764-765, 773, 793 and 804 in Figure legends:

Data shown are representative of at least three independent experiments.

Reviewer 2's comment 15: Throughout the paper, the authors need to refer to specific panels (e.g., Fig. 4A) and not only to figures as a whole (e.g., Fig. 4) when presenting or discussing specific results.

Authors' responses: As pointed out by Reviewer 2, we indicated specified panels in the text.

Reviewer 2's comment 16: The Introduction is somewhat unfocused. First, the long first paragraph goes back and forth between *Aspergillus* and *Mucorales*. Second, given that this is a predominantly immunologic/mechanistic paper, there is no need to spend 21 lines on the ineffectiveness of antifungal therapies and need for adjunct immunotherapies. This point can be made much more concisely, as emerging resistance and limitations of antifungal therapy are well known to the readership of *Microbiology Spectrum*. Third, the rationale for antifungal immunotherapy is not only based on the limitations of antifungal therapy but on the fact that outcomes of invasive mold infections in severely immunocompromised patients are predominantly host driven (i.e., remission of underlying hematological malignancies and recovery of cytopenia). There is abundant literature on this phenomenon, especially from U.S. cancer centers. Fourth, the properties and known functions of

$\gamma\delta$ T cells, especially in the context of infectious diseases, should be introduced more comprehensively. To that end, I would suggest putting the content of lines 266-284 in the Introduction.

Authors' responses: We revised the Introduction to address the comments raised by Reviewer 2. We moved the texts from the Discussion to the Introduction. The revised version of the Introduction is as follows:

Lines 66-112:

Aspergillus molds, especially *Aspergillus fumigatus*, are causative agents of invasive aspergillosis (IA), a fatal disease occurring mainly in severely immunocompromised hosts, such as patients with hematologic malignancies and transplant recipients (1–3). Despite the development of novel antifungal agents and improved treatment strategies, the mortality rate of IA remains high at 35% (1). Furthermore, the increase and global spread of azole-resistant *A. fumigatus* hampers conventional IA treatments and poses a worldwide challenge (4–7). In addition, mucormycosis is an invasive fungal infection caused by over 200 species, including *Rhizopus spp* and *Cunninghamella spp* (8). These fungal species are naturally resistant to azole and other antifungals, contributing to a high fatality rate of 54% (9).

Recent research advances in the mechanism underlying immune responses to *Aspergillus spp.* (10–12) indicate that IA resistance to standard therapy may result from host's failure to induce appropriate immune responses. Indeed, the outcome of invasive mold infections in severely immunocompromised patients depends on host factors, including the resolution of neutropenia (13, 14). This has led to the emergence of immunotherapy as an alternative approach to conventional antifungal drug treatment (15). The antifungal effects of natural killer (NK) cells against *A. fumigatus* have been demonstrated, and the adoptive transfer of NK cells for treating IA has been explored (16–18).

Besides NK cells, innate or innate-like immune effector cells exist in humans, such as CD4CD8-double negative T cells, including NKT cells and $\gamma\delta$ T cells. Human V γ 9V δ 2-bearing $\gamma\delta$ T cells (V γ 9V δ 2 T cells) account for 1–5% of circulating T cells in the peripheral blood, exhibit innate immune-like functions, and can damage infected cells and tumor cells similar to NK cells (19,20). In a previous study, V γ 9V δ 2 T cells produced a significant level of tumor necrosis factor- α (TNF- α) in response to water-soluble extracts from *Aspergillus spp.* (21); however, the antigen and mechanism underlying the antigen recognition remain unknown. It is also unknown whether V γ 9V δ 2T cells possess antifungal activity against *Aspergillus*.

In healthy adults, the majority of peripheral blood $\gamma\delta$ T cells express V γ 9V δ 2-bearing TCRs, which recognize microbial (*E*)-4-hydroxy-3-methyl-but-2-enyl pyrophosphate (HMBPP) in

the 2-C-methyl-D-erythritol 4-phosphate/1-deoxy-D-xylulose 5-phosphate (MEP/DOXP) pathway and self-isopentenyl diphosphate (IPP) and dimethylallyl diphosphate (DMAPP) in the mevalonate pathway in a butyrophilin (BTN) 2A1/3A1-dependent manner (22, 23).

HMBPP is produced by some pathogenic bacteria, such as *Mycobacterium tuberculosis*, *Mycobacterium bovis*, *Listeria monocytogenes*, *Escherichia coli*, *Salmonella typhimurium*, and parasites, such as *Plasmodium falciparum* and *Toxoplasma gondii* (24, 25). When V γ 9V δ 2 T cells recognize these pathogen-derived metabolites, they readily proliferate and produce interferon- γ (IFN- γ) and TNF- α (26), mounting a rapid response against the pathogens. The antibacterial activity of V γ 9V δ 2 T cells against *M. tuberculosis* has been reported (27, 28). A clinical study on allogeneic V γ 9V δ 2 T cell-based immunotherapy in patients with multidrug-resistant tuberculosis demonstrated the regimen to be well tolerated and effective against the pathogen (29), indicating its potential applicability to IA.

We have previously shown that V γ 9V δ 2 T cells from peripheral blood could be expanded ex-vivo using PTA, a nitrogen-containing bisphosphonate prodrug and an inhibitor of farnesyl diphosphate synthase (FDPS), and interleukin-2 (IL-2) (30, 31). In addition, a clinical trial of therapeutic administration of V γ 9V δ 2 T cells to patients with malignant tumors has revealed that the regimen is well tolerated (32). In this study, we evaluated the antifungal activity of ex-vivo expanded/activated human V γ 9V δ 2 T cells against *Aspergillus spp.* and other *Mucorales* in vitro and explored the mechanism underlying their antifungal activity. Our findings will help develop V γ 9V δ 2 T cells as a novel treatment modality for IA or mucormycosis.

Reviewer 2's comment 17: Several statements in the Discussion, including references to the authors' own prior work, require references, e.g., in lines 266-273 or lines 285-295.

Authors' responses: As pointed out by Reviewer 2, we added references to the sentences (lines 91-95 and 280-290).

Reviewer 2's comment 18: The Materials & Methods section is overly detailed, especially with regard to standard methodology (e.g., isolation of PBMCs). These sections might be shortened or relegated to a Supplementary Methods section.

Authors' responses: As suggested by Reviewer 2, only brief descriptions regarding the derivation of V γ 9V δ 2 T cells were left in the Materials and Methods of the manuscript as follows, and more detailed descriptions were moved to Supplementary Methods.

Lines 387-390 in the Materials and Methods:

Human PBMC were obtained from healthy adult volunteers. V γ 9V δ 2 T cells were derived by stimulating PBMC with PTA and IL-2 for 11 days. NK cells were ex-vivo expanded by treating CD3-depleted PBMC with IL-2/IL-18 for 10 days. Detailed methods for V γ 9V δ 2 T cells and NK cells expansion are described in Supplementary Methods.

Reviewer 2's comment 19: The manuscript is overall well-written but some editing would be needed to correct minor grammar mistakes and stylistic issues.

Authors' responses: Several authors reviewed the manuscript and grammatical errors and typos were removed, as pointed out by Reviewer 2.

In this manuscript, we examined the effect of $\gamma\delta$ T cells on *Apergillus* and fungi and determined the in-vitro antifungal activity of the T cell subset. We believe that the present findings pave the road to the establishment of immunotherapy for invasive fungal diseases.

Sincerely,

Takahiro Takazono, M.D., Ph.D.
Department of Respiratory Medicine,
Graduate School of Biomedical Sciences,
Nagasaki University,
1-7-1 Sakamoto, Nagasaki 852-8501, Japan

Re: Spectrum03614-23R1 (Human V γ 9V δ 2 T cells exhibit antifungal activity against *Aspergillus fumigatus* and other filamentous fungi)

Dear Dr. Takahiro Takazono:

Thank you for the privilege of reviewing your work. Below you will find my comments and instructions from the Spectrum editorial office.

I appreciate you revising the manuscript and thoroughly addressing the reviewer's comments. I suggest the following editorial changes before publication.

Throughout the manuscript, I suggest that you use the terms "susceptible" or "resistant" when referring to drug susceptibility (i.e., to voriconazole). Please use "antifungal activity" or "antifungal effect" when referring to the T cells.

Lines 172-174: "Aspergillus hyphae formed hyphal layers with numerous mycelia intertwined on a polystyrene plate, and the mycelia within the structures were more resistant to antifungal agents including voriconazole than planktonic ones (33, 34)". I suggest clarifying that this does not refer to your experiments. Alternatively, you can move the sentence to the Discussion.

Please clarify the following (lines 179-180): "All species of *A. fumigatus*, including azole-susceptible and several azole-resistant strains, were resistant to voriconazole."

Please also rephrase the sentence in lines 305-308. You can state that your experiments indicate that the antifungal activity of T cells is not affected by drug resistance mutations (as anticipated). Remove "might have nothing to do".

Line 42 "some cases of mucormycosis", line 273 "some Mucorales", line 370 "some species of Mucorales". Avoid the term "some".

As the above changes are minor, please return the manuscript within 20 days. If you cannot complete the modification within this time period, please contact me. If you do not wish to modify the manuscript and prefer to submit it to another journal, notify me immediately so that the manuscript may be formally withdrawn from consideration by Spectrum.

Revision Guidelines

Sincerely,
Paschalis Vergidis
Editor
Microbiology Spectrum

January 29, 2024

Dear Editor,

The authors thank the Reviewers for their valuable comments. We have now revised our manuscript to address these comments with point-by-point responses.

Reviewer's comment 1:

Throughout the manuscript, I suggest that you use the terms "susceptible" or "resistant" when referring to drug susceptibility (i.e., to voriconazole). Please use "antifungal activity" or "antifungal effect" when referring to the T cells.

Authors' responses:

Thank you for your suggestions. We modified several sentences according to your suggestion. Final versions are as below.

Line 170-173 in the Results:

When azole-resistant *A. fumigatus* strains (hyphae-form) were co-cultured with V γ 9V δ 2 T cells, the V γ 9V δ 2 T cells exhibited potent antifungal activity against these fungi to a similar level as with the azole-susceptible *A. fumigatus* strain (Fig. 2).

Line 226-227 in the Results:

Based on the aforementioned findings, it is evident that V γ 9V δ 2 T cells could exert antifungal effects.

Line 302-305 in the Discussions:

In addition, V γ 9V δ 2 T cells also exhibited antifungal effects against some azole-resistant *A. fumigatus* strains with different resistance mechanisms (NGS-ER7, MF2108, IFM63240), strongly indicating that V γ 9V δ 2 T cell-based immunotherapy could serve as an alternative modality for the treatment of IA caused by wild-type and azole-resistant *A. fumigatus*.

Line 321-324 in the Discussions:

In this study, we used an *A. fumigatus* mutant strain, Δ uge5, which is defective in GM, and found that V γ 9V δ 2 T cells exhibited antifungal effects against GM-deficient *A. fumigatus* as the wild type *A. fumigatus* (Af293).

Reviewer's comment 2:

Lines 172–174: “*Aspergillus* hyphae formed hyphal layers with numerous mycelia intertwined on a polystyrene plate, and the mycelia within the structures were more resistant to antifungal agents including voriconazole than planktonic ones (33, 34)”. I suggest clarifying that this does not refer to your experiments. Alternatively, you can move the sentence to the Discussion.

Authors' responses:

Thank you for your suggestions. We modified the sentence you pointed out as below, and move that sentence from the Results part to Discussions part.

Line 310-312 in the Discussions:

Notably, it was reported that *Aspergillus* hyphae formed hyphal layers with numerous mycelia intertwined on a polystyrene plate, and the mycelia within the structures were more resistant to antifungal agents including voriconazole than planktonic ones (44, 45).

Reviewer's comment 3:

Please clarify the following (lines 179–180): “All species of *A. fumigatus*, including azole-susceptible and several azole-resistant strains, were resistant to voriconazole.”

Authors' responses:

Thank you for your suggestions. We modified the sentence as below.

Line 177-179 in the Results:

Alternatively, all species of *A. fumigatus*, including azole-susceptible and several azole-resistant strains, were resistant to voriconazole on the condition of forming the hyphal layer (Fig. 2).

Reviewer's comment 4:

Please also rephrase the sentence in lines 305–308. You can state that your experiments indicate that the antifungal activity of T cells is not affected by drug resistance mutations (as anticipated). Remove “might have nothing to do”.

Authors' responses:

Thank you for your suggestions. We modified the sentence as below.

Line 305-308 in the Discussions:

Since common mutations in *A. fumigatus* responsible for drug resistance are generally unrelated to the structure and components of cell walls that are highly antigenic for the host immune system, the antifungal activity of V γ 9V δ 2 T cells is not affected by such azole-resistant mutations.

Reviewer's comment 5:

Line 42 "some cases of mucormycosis", line 273 "some Mucorales", line 370 "some species of Mucorales". Avoid the term "some".

Authors' responses:

Thank you for your suggestions. The term "some" has been deleted in the sentence you pointed out, and for the sentences line 269-271 and line 371-373, the specific name of the species has been added instead. The final version is as follows.

Line 36-38 in the Abstract

In addition, V γ 9V δ 2 T cells exhibited antifungal activity against hyphae of all *Aspergillus spp.*, *Cunninghamella bertholletiae* and *Rhizopus microsporus*, but not against their conidia.

Line 42-43 in the Abstract:

V γ 9V δ 2 T cells could be developed as a novel modality for treating IA or mucormycosis.

Line 270-272 in the Discussions:

V γ 9V δ 2 T cells exhibited significant antifungal effects against many species of filamentous fungi-hyphae, including azole-susceptible and resistant *A. fumigatus*, and *Mucorales* such as *Cunninghamella bertholletiae* and *Rhizopus microsporus*.

Line 372-374 in the Discussions:

Second, although it was indicated that GM itself was not recognized by V γ 9V δ 2 T cells, the antigenic entity of *A. fumigatus* and *Mucorales* such as *Cunninghamella bertholletiae* and *Rhizopus microsporus* for V γ 9V δ 2 T cells remains unidentified.

In addition, we realize that *Rhizopus* microspores was consistently misspelled in the manuscript, then, we corrected them.

Sincerely,

Takahiro Takazono, M.D., Ph.D.
Department of Respiratory Medicine,
Graduate School of Biomedical Sciences,
Nagasaki University,
1-7-1 Sakamoto, Nagasaki 852-8501, Japan

Re: Spectrum03614-23R2 (Human V γ 9V δ 2 T cells exhibit antifungal activity against *Aspergillus fumigatus* and other filamentous fungi)

Dear Dr. Takahiro Takazono:

Your manuscript has been accepted, and I am forwarding it to the ASM production staff for publication. Your paper will first be checked to make sure all elements meet the technical requirements. ASM staff will contact you if anything needs to be revised before copyediting and production can begin. Otherwise, you will be notified when your proofs are ready to be viewed.

Sincerely,
Paschalis Vergidis
Editor
Microbiology Spectrum